# Daily reference evapotranspiration prediction for irrigation scheduling decisions based on the hybrid PSO-LSTM model

**Weibing Jia** [1]○, **Yubin Zhang**[1,2]○, **Zhengying Wei**[1] *, **Zhenhao Zheng**[3], **Peijun Xie**[2]

**1** The State Key Laboratory for Manufacturing Systems Engineering, Xi'an Jiaotong University, Xi'an, Shaanxi, China, **2** College of Digital Technology and Engineering, Ningbo University of Finance and Economics, Ningbo, Zhejiang, China, **3** Agricultural Technology Extension Station of Ningbo, Ningbo, Zhengjiang, China

○ These authors contributed equally to this work.
* zywei@mail.xjtu.edu.cn

**Data Availability Statement:** All relevant data are within the paper and its Supporting Information files.

## Abstract

The shortage of available water resources and climate change are major factors affecting agricultural irrigation. In order to improve the irrigation water use efficiency, it is necessary to predict the water requirements for crops in advance. Reference evapotranspiration ($ET_o$) is a hypothetical standard reference crop evapotranspiration, many types of artificial intelligence models have been applied to predict $ET_o$; However, there are still few in the literature regarding the application of hybrid models for deep learning model parameters optimization. This paper proposes two hybrid models based on particle swarm optimization (PSO) and long-short-term memory (LSTM) neural network, used to predict $ET_o$ at the four climate stations, Shaanxi province, China. These two hybrid models were trained using 40 years of historical data, and the PSO was used to optimize the hyperparameters in the LSTM network. We applied the optimized model to predict the daily $ET_o$ in 2019 under different datasets, the result showed that the optimized model has good prediction accuracy. The optimized hybrid models can help farmers and irrigation planners to make plan earlier and precisely, and can provide valuable information to improve tasks such as irrigation planning.

## Introduction

Reference evapotranspiration is a hypothetical standard reference crop evapotranspiration, which plays a broad and important role in irrigation decision-making, hydrological prediction and scheduling, crops growth simulation and climate disaster monitoring [1]. The $ET_o$ can be determined using lysimeter, which provide accurate measurements typically using in the development and validation of other methods. Given the cost and complexity of lysimeters, its use is typically restricted to research. Thus, the use of mathematical models based on meteorological data (temperature, relative humidity, solar radiation and wind speed) recorded by weather stations is a more suitable approach for practical applications [2]. The Penman-Monteith FAO-56 method is used as the standard method to estimate the $ET_o$, and this method has served as a criterion for comparing the forecast values of other models [3].

**Funding:** This work was supported in part by the National Key Research and Development Project of the 13th five-year plan fertilizer-water source-equipment adaptation technology and control equipment (No.2017YFD0201504)(http://www.most.gov.cn/index.html), by the Key R&D Program of Shaanxi Province (2023-YBNY-202), by Ningbo Science and Technology Plan Project (2021S022), by the Zhejiang Province Basic Public welfare Research Program (LGN20F030001), and by the Key Industrial Innovation Chain Projects of Shaaxi Province (2023-ZDLNY-67).

**Competing interests:** The authors have declared that no competing interests exist.

Over the last few years, with the rapid development of computers' computing ability and artificial intelligence theory, computing $ET_o$ using weather data has been considered a regression task that can be solved by some classical machine learning model for estimating $ET_o$ [4–6], different types of artificial intelligence methods have been applied to estimate and predict $ET_o$, including ANN, gene expression programming (GEP), support vector machine (SVM), random forest (RF), extreme gradient boosting (XGBoost), adaptive neuro-fuzzy inference system (ANFIS), and multi-layer perceptron neural network [7–11]. The deep learning techniques such as the Temporal Convolution Network, the Convolutional Neural Networks (CNN), the Long Short-Term Memory model (LSTM) and hybrid the Convolutional Neural Networks and the Long Short-Term Memory model (CNN-LSTM) also have been recently employed to predict $ET_o$, and these models show outstanding performances [12–16]. Although these models have been used to estimate daily or monthly $ET_o$, estimation studies of model hyper parameters are not as common in the literature as $ET_o$ forecasting studies. The decision making in irrigation scheduling depends on forecasts of 1 to 10 days ahead, which is critically important to determine crop water requirements and real time irrigation scheduling. The shorter time from daily to 7 days can be useful in planning the use of irrigation systems as well as in optimizing system power consumption. Longer forecast horizons also help in the water management of irrigation channels and reservoirs for irrigation use [17, 18].

In recent year, the LSTM and CNN are probably the most popular, efficient and widely used deep techniques for time series forecasting. LSTMs were introduced to overcome the problems of vanishing gradients of RNNs with capability of storing important information containing long sequences, which were used in a number of applications, including speech recognition, stock price prediction, image text recognition, traffic flow forecasting, agriculture rainfall forecast, and grammar learning [19, 20]. However, the application of LSTM in the field of hydrology has not been widely reported in the literature, but can be used in the estimation of hydrologic variables since several climatic variables used in hydrology exhibit time series behavior [21]. In the LSTM neural network, since the model training is an automatic process of adjusting weights and thresholds, the values of the hyperparameters will directly affect the occurrence of convergence, learning time, and local minima. However, in the existing study, the LSTM models provided only slight performance gains. Furthermore, the models generally have more hyperparameters to be adjusted, requiring more intelligent algorithm to optimize them. Hybrid techniques, such as ensemble modeling, usually offer better results than simple techniques since the combination of the models tends to capture the best of each one. Hybrid models are also developed to combine the advantages of different methods and form a new forecasting strategy. Therefore, many of them are considered to be more effective than pure classical methods or artificial intelligence models [22]. Particle swarm optimization (PSO) is a global optimization algorithm with simple rules and fast convergence. It has been widely used in neural network training and structural optimization design [23]. However, most parameters in the LSTM model need to be set manually, which is inefficient and unreasonable. The potential of a hybrid algorithm based on LSTM and PSO is still unexplored in the literature for $ET_o$ forecasting. Considering the optimization of the parameters of the shortcomings of the LSTM model, the most recently advanced hybrid technique was selected to improve the performance of the LSTM forecasting model. Specifically, the PSO was used to optimize the hyperparameters in the LSTM network, and proposes two hybrid model used to predict the daily $ET_o$ in four climate stations, Shaanxi province, China. Compared to existing models, the optimized model has better feasibility and prediction accuracy.

The innovation of this paper lies mainly in the following: two hybrid deep neural network model were proposed for daily $ET_o$ forecasting. The number of hidden neurons, dropout and look-back in LSTM are optimized by PSO. The optimized models were tested with four

different datasets, and the proposed models achieved good results in forecast accuracy under different datasets.

This paper is organized as follows, in material and methods, an overview of study area, reference evapotranspiration and LSTM prediction model were given, and two hybrid models based on PSO and LSTM were introduced. In results and discussion, the result and analysis of optimized forecasting model were given, and we compare the proposed model under four different datasets. In conclusions, we conclude this paper with future work.

## Material and methods

### Study site

The Guanzhong Basin is a typical and important grain producing area in China, is located in the central part of Shaanxi province in China, and covers an area of 20 440 km$^2$ (33˚39′ - 35˚ 50′ N, 107˚30′ -110˚37′ E), it has a warm temperate semi-humid monsoon climate, with the average annual temperature of 13.7˚ C, and the mean annual precipitation recorded from 2000 to 2010 was approximately 615 mm. The Wugong (WG, 34˚19′ N, 108˚14′E, 471.0m), Feng-xiang (FX, 34˚31′ N, 107˚23′E, 781.1m), Xianyang (XY, 34˚24′ N, 108˚43′E, 472.8m) and Pucheng (PC, 34˚53′ N, 109˚38′E, 387.2m) weather stations are located in the Guanzhong Basin [24, 25], as the study sites for ET$_o$ forecasting.

### Reference evapotranspiration

The general FAO Penman-Monteith approach (FAO56) [26] has been widely used to calculate reference evapotranspiration, because most users appreciate its simplicity and consider that it has acceptable accuracy. The FAO56 is expressed as Eq (1):

$$ET_o = \frac{0.408\Delta(Rn - G) + \gamma\frac{900}{T+273}\mu_2(e_s - e_a)}{\Delta + \gamma(1 + 0.34\mu_2)} \tag{1}$$

Herein, ET$_o$ represents the reference evapotranspiration (mm day$^{-1}$); R$_n$ represents net radiation at the crop surface (MJ m$^{-2}$ day$^{-1}$); G represents soil heat flux density (MJ m$^{-2}$day$^{-1}$); T represents mean daily air temperature at 2 m height (˚C); u$_2$ represents wind speed at 2 m height (m s$^{-1}$); e$_s$ represents saturation vapor pressure (kPa); e$_a$ represents actual vapor pressure (kPa); (e$_s$ – e$_a$) represents saturation vapor pressure deficit (kPa); Δ represents slope vapor pressure curve (kPa ˚C$^{-1}$); γ represents psychrometric constant (kPa ˚ C$^{-1}$).

The China National Meteorological Data Center (http://data.cma.cn, accessed on 30 December 2019) provides accurate and free weather data, the weather data types include daily mean temperature (T$_{mean}$,˚C), daily maximum temperature (T$_{max}$,˚C), daily minimum temperature (T$_{min}$,˚C), average relative humidity (RH, %), daily average wind speed (U$_2$, m/s) and sunshine hours (S$_h$, h). From January 1, 1980 to December 31, 2019, a total of 14600 weather data sample were obtained for each weather station (accessed on December 31, 2019). Table 1 presents the data statistics for each weather parameters. Since the elevations and latitude of the four stations are similar, the maximum, average and minimum values of the six weather parameters at the four stations are close to each other. It can be inferred that a predictive model optimized using weather data from one of the four stations can be used to predict parameters for the other sites.

Using weather data and the FAO56 approach, a total of 14600 daily ET$_o$ data were obtained for each weather station (Fig 1). It can be seen from the figure the value ranges for ET$_o$ are from 0.3 mm to 10 mm, and the peak period for each year's data is from July to September. Due to cyclical changes in weather parameters, the data of ET$_o$ fluctuate greatly in different

**Table 1. Data descriptive statistics for the period 1980–2019.**

| Site | Parameter | $T_{max}$ | $T_{mean}$ | $T_{min}$ | RH | $U_2$ | $S_h$ | $ET_o$ |
|------|-----------|-----------|------------|-----------|-----|-------|-------|--------|
| WG | Max | 41.0 | 34.0 | 28.6 | 100 | 7.5 | 13.4 | 8.53 |
| | Average | 19.3 | 13.6 | 9.1 | 70.9 | 1.4 | 4.8 | 2.47 |
| | Min | -4.9 | -11.9 | -17.7 | 4.5 | 0.0 | 0.0 | 0.37 |
| | Standard deviation | 10.21 | 9.64 | 9.38 | 14.96 | 0.79 | 4.17 | 1.62 |
| FX | Max | 40.5 | 32.3 | 26.9 | 100 | 9.0 | 13.7 | 8.76 |
| | Average | 17.7 | 12.0 | 7.41 | 68.9 | 1.93 | 5.3 | 2.54 |
| | Min | -7.2 | -14.1 | -19.2 | 15.0 | 0.0 | 0.0 | 0.34 |
| | Standard deviation | 10.05 | 9.47 | 9.13 | 16.53 | 0.82 | 4.24 | 1.62 |
| XY | Max | 41.7 | 34.8 | 29.8 | 100 | 12.3 | 13.8 | 9.64 |
| | Average | 19.2 | 13.4 | 8.5 | 69.6 | 2.1 | 5.5 | 2.72 |
| | Min | -6.1 | -12.6 | -18.6 | 14.0 | 0.0 | 0.0 | 0.31 |
| | Standard deviation | 10.46 | 9.99 | 9.81 | 15.63 | 1.25 | 4.33 | 1.81 |
| PC | Max | 41.4 | 35.5 | 29.6 | 100 | 10.3 | 14.0 | 9.27 |
| | Average | 19.6 | 14.0 | 9.5 | 61.9 | 2.1 | 6.03 | 2.94 |
| | Min | -7.0 | -12.5 | -16.9 | 10.0 | 0.0 | 0.0 | 0.33 |
| | Standard deviation | 10.51 | 10.11 | 9.85 | 18.59 | 1.02 | 4.25 | 1.85 |

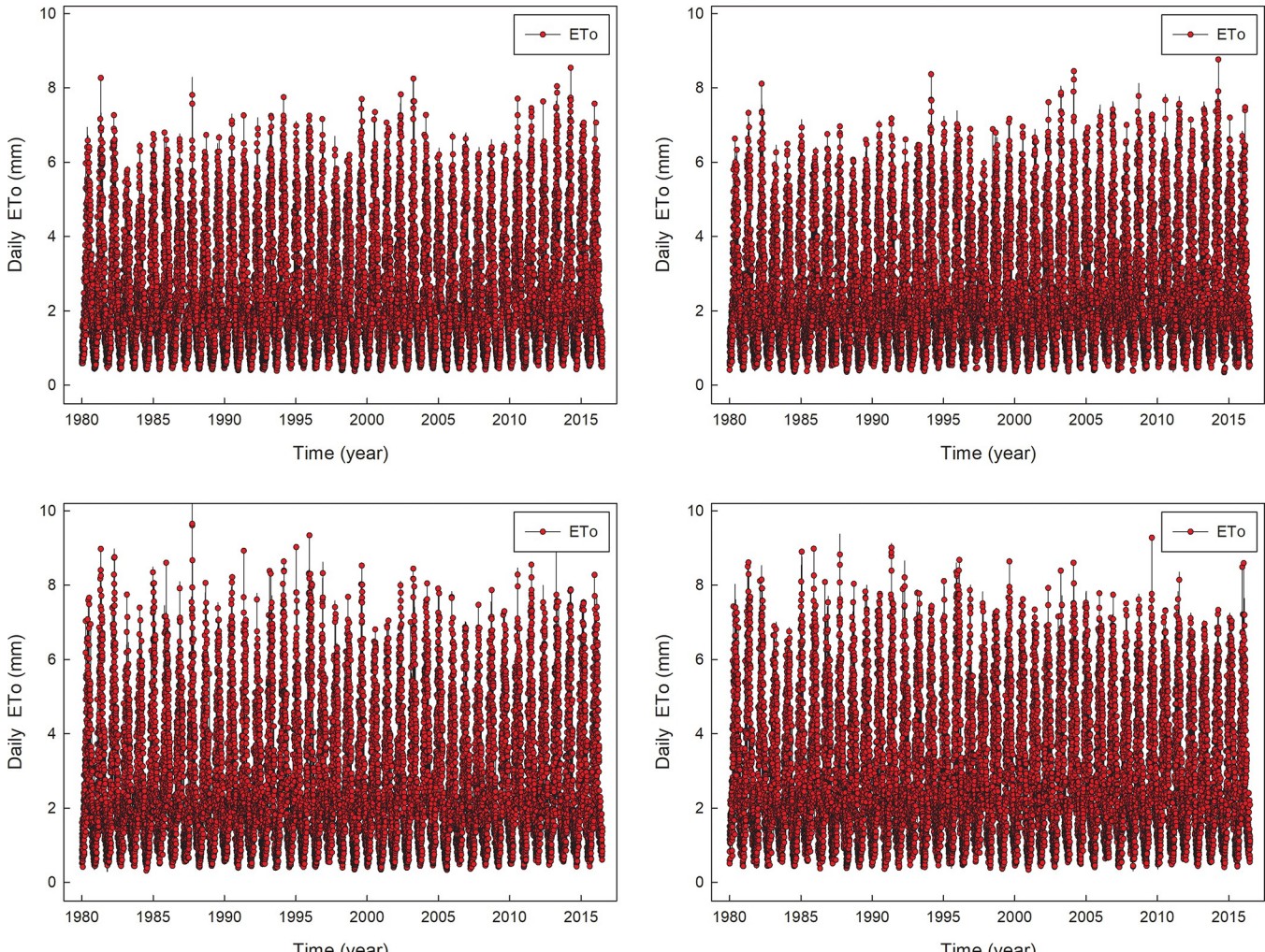

**Fig 1. Timing diagram of daily $ET_o$ at four weather stations.**

days, $ET_o$ changes with periodic changes in weather parameters. Table 1 presents the data statistics for the weather data and $ET_o$.

## Long short-term memory network model

The LSTM are widely applied in time series forecasting, which consists of input layer, LSTMs layer and output layer. The main components of the LSTM network consist of a sequence input layer that is used to input a sequence (time series data), and a sequence output layer that is used to learn long-term reliance among the time steps of a sequence data [27].

An LSTM-NN cell consists of three gates: input gate, output gate and forget gate. The framework of LSTM cell is shown in (Fig 2), the input gate gives new input to the cell. The output gate specifies the output of the cell, and the forget gate is responsible for specifying the prior values that need to be retained for future reference. The work principle of LSTM neural network at time step t are as follows [27, 28]:

$$f(t) = \sigma(W_f \cdot [h_{t-1}, x_t] + b_f) \tag{2}$$

$$i(t) = \sigma(W_i \cdot [h_{t-1}, x_t] + b_i) \tag{3}$$

$$\tilde{C}(t) = tanh(W_c \cdot [h_{t-1}, x_t] + b_c) \tag{4}$$

$$C(t) = f_t * C_{t-1} + i_t * \tilde{C}_t \tag{5}$$

$$O_t = \sigma(W_O \cdot [h_{t-1}, x_t] + b_O) \tag{6}$$

$$h(t) = O_t * tanh(C_t) \tag{7}$$

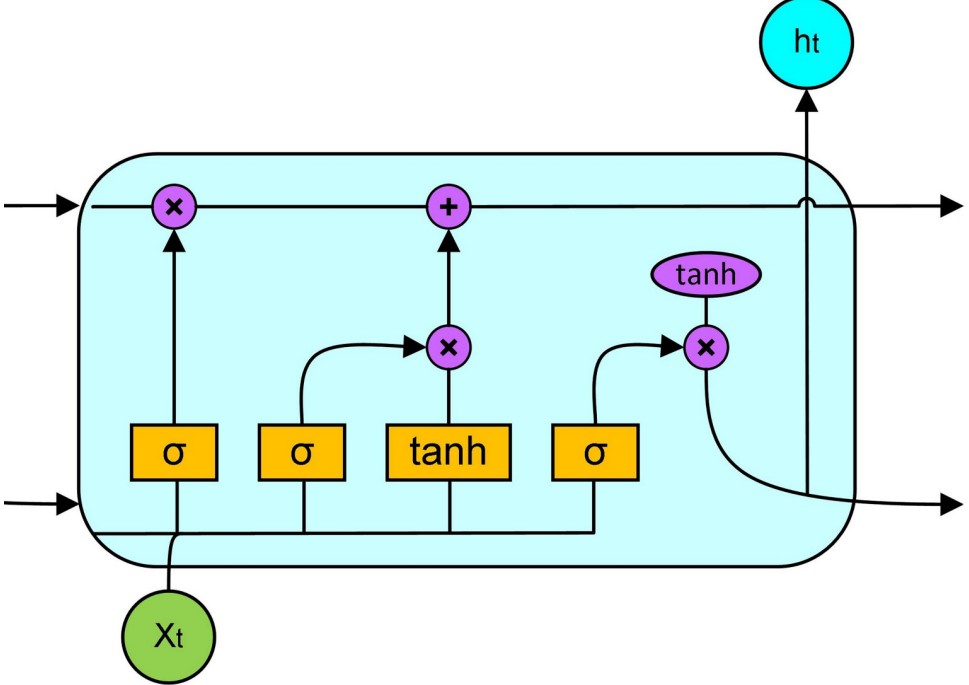

**Fig 2. The basic LSTM network architecture for regression problems.**

$$sigmoid(x) = 1/(1 + e^{-x}) \tag{8}$$

$$\tanh(x) = (e^x - e^{-x})/(e^x + e^{-x}) \tag{9}$$

The $\sigma$ refers to the sigmoid function and controls the information passing state. When the $\sigma$ is 0, the nothing can pass. When the $\sigma$ is 1, everything can pass. $W_f$, $W_i$, $W_c$ and $Wo$ refer to the input weight. The corresponding $b_f$, $b_i$, $b_c$ and $b_o$ refer to the biasing. The t and t-1 refer to the current and previous time status. The x and h refer to the input and output, and C refers to the cell status.

## Particle swarm optimization algorithm

As one of the evolutionary calculation technologies, the PSO algorithm searches an optimal solution of each particle in each iteration and records it as the current individual extremum (particle best, pbest), compares all current individual in search space, the best is denoted as the global extremum (global best, gbest) of the entire particle swarm.

Each particle has its speed and position in the next each subsequent iteration, all the particles in the particle swarm adjust their speed and position by pbest and gbest [29]. The particle updates its speed and position according to the following Formulas (10) and (11) [30]:

$$v_{id}^k = wv_{id}^{k-1} + c_1 r_1(pbest_{id} - x_{id}^{k-1}) + c_2 r_2(gbest_d - x_{id}^{k-1}) \tag{10}$$

$$x_{id}^k = x_{id}^{k-1} + v_{id}^{k-1} \tag{11}$$

where k denotes the number of iterations. $c_1$ and $c_2$ are acceleration constant, which are used to adjust the maximum learning step. w is the inertia factor, which is used to adjust the search range of solution space, and $r_1$ and $r_2$ are uniform random numbers within the range [0, 1] to increase the randomness of the search.

## Hybrid model based on LSTM and PSO

Similar to other neural networks, the LSTM is very sensitive to hyperparameters, including the size of the time window, the batches, hidden layer neurons, dropout, activate function.

In this study, the framework of two PSO-LSTM model is constructed, as shown in (Fig 3). the first model consisting of a LSTM layer, a flatten layer and a dense layer, and two important hyperparameters, the number of the first hidden layer neurons ($N_1$) and the look back (time windows, $T_1$) were used to be optimized. The second model consisting of two LSTM layer, a flatten layer and a dense layer, and four important hyperparameters, the number of the first hidden layer neurons ($N_2$), the second hidden layer neurons ($N_3$), the dropout (D) and the look back ($T_2$) were used to be optimized. Those hyperparameters are regarded as the particles in PSO. The mean absolute error (MAE) of the model's predicted and actual values are taken as the fitness function.

The implementation process of the hybrid model was shown in (Fig 4). The detailed optimized process of the hybrid model is as follows [29, 31]:

**Step 1**: Import the dataset and normalization, and dividing the dataset into training set, validation set and test set.

**Step 2**: PSO is used to optimize the hyperparameters.

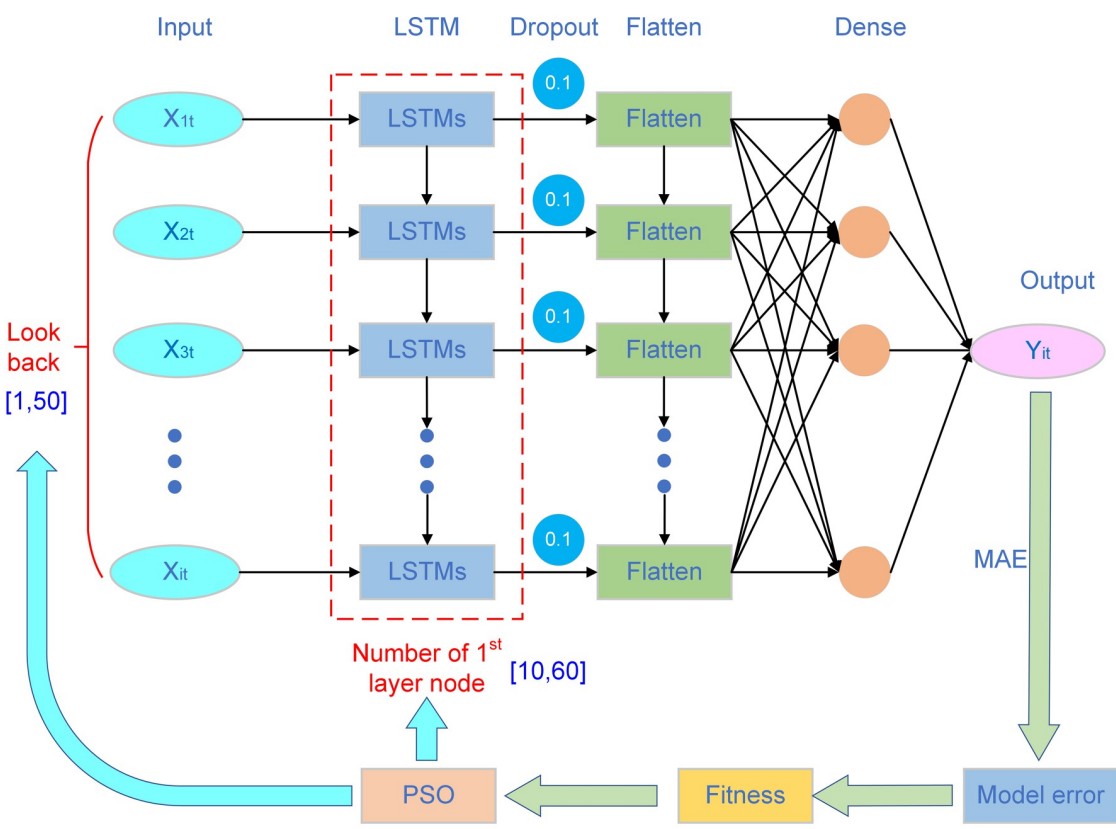

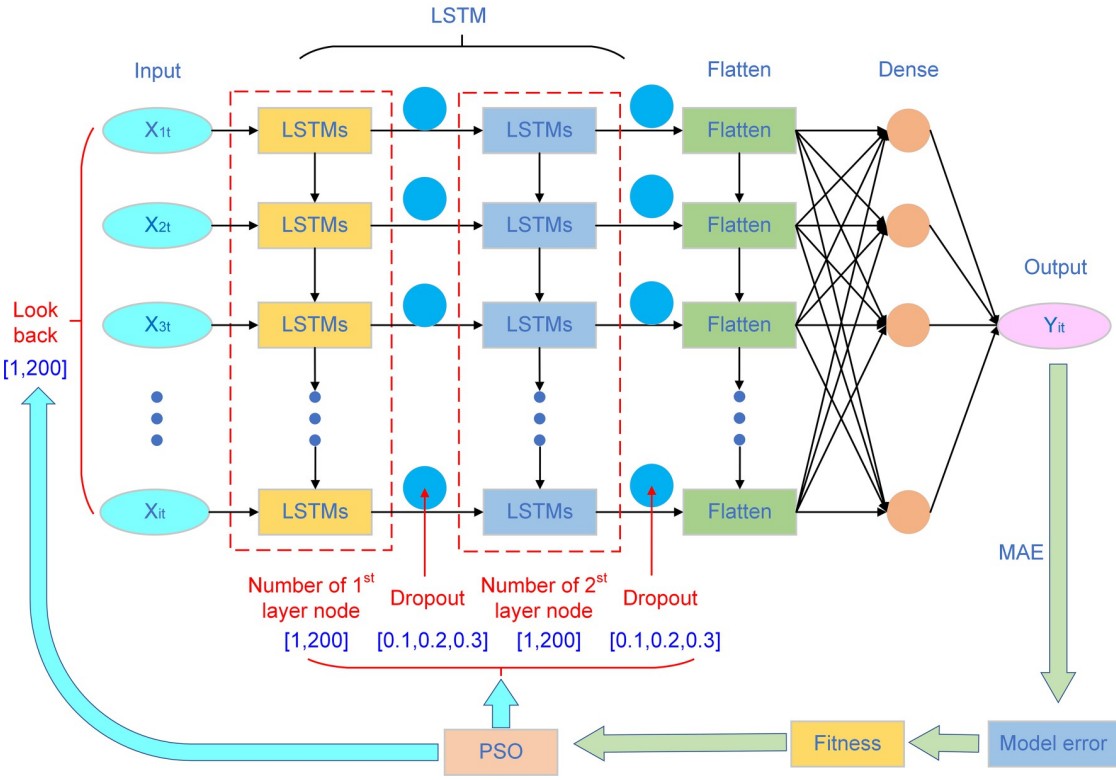

**Fig 3. The framework of the two PSO-LSTM models.**

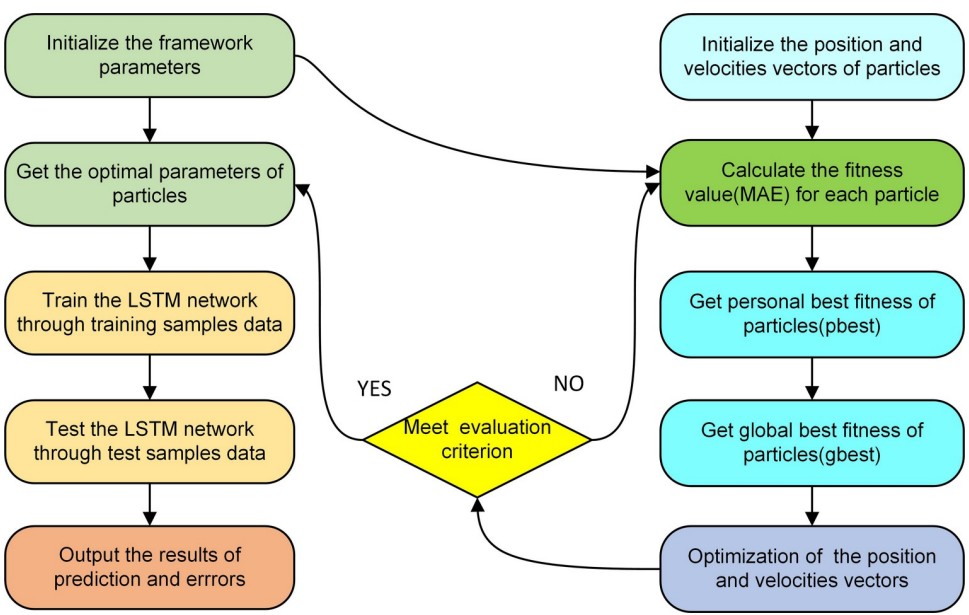

**Fig 4. The implementation process of the PSO-LSTM model.**

- Parameter initialization. The particle dimension, population size, iterations, learning factors $c_1$ and $c_2$, inertia weight w, velocity and position are determined.

- Initialize the velocity and position of particles, and randomly generate population particles.

- The mean absolute error is taken as the fitness value of the model, and the pbest and gbest are calculated with the fitness value of the particle.

- After each iteration, the position and velocity are updated, and the fitness is also calculated, at the same time, the pbest and gbest are also updated.

- Determine whether the termination condition is met. If satisfied, the optimization results were obtained. Otherwise, go back to step 3).

    **Step 3:** The optimized LSTM model were obtained, and training and evaluate the optimized model with other weather stations data.

## Data pre-processing

The stationarity of daily $ET_o$ is important for LSTM model forecasting, long-term observational $ET_o$ records are needed to detect the stationarity. The detection method for the sequence data stationarity includes the intuitive judgment of sequence data image, the method of the Dickey-Fuller test [32], the method of the autocorrelation coefficient diagram and the partial correlation coefficient diagram. The partial autocorrelation coefficient (PAC) of the sequence data represents the correlation between any two different time steps for the same time series. The PAC of daily $ET_o$ was shown in (Fig 5), and show the lags from 0 to 50 days. It can be seen from the figure that the PAC of daily $ET_o$ has a rapid decay to near zero with increases of the lag numbers, the confidence limits for the autocorrelations were ±0.02 at the 95% confidence intervals. Although the PAC of daily $ET_o$ changes when the lag is greater than 10, the fluctuation range is little, so the sequence data of daily $ET_o$ was a stationary random parameter.

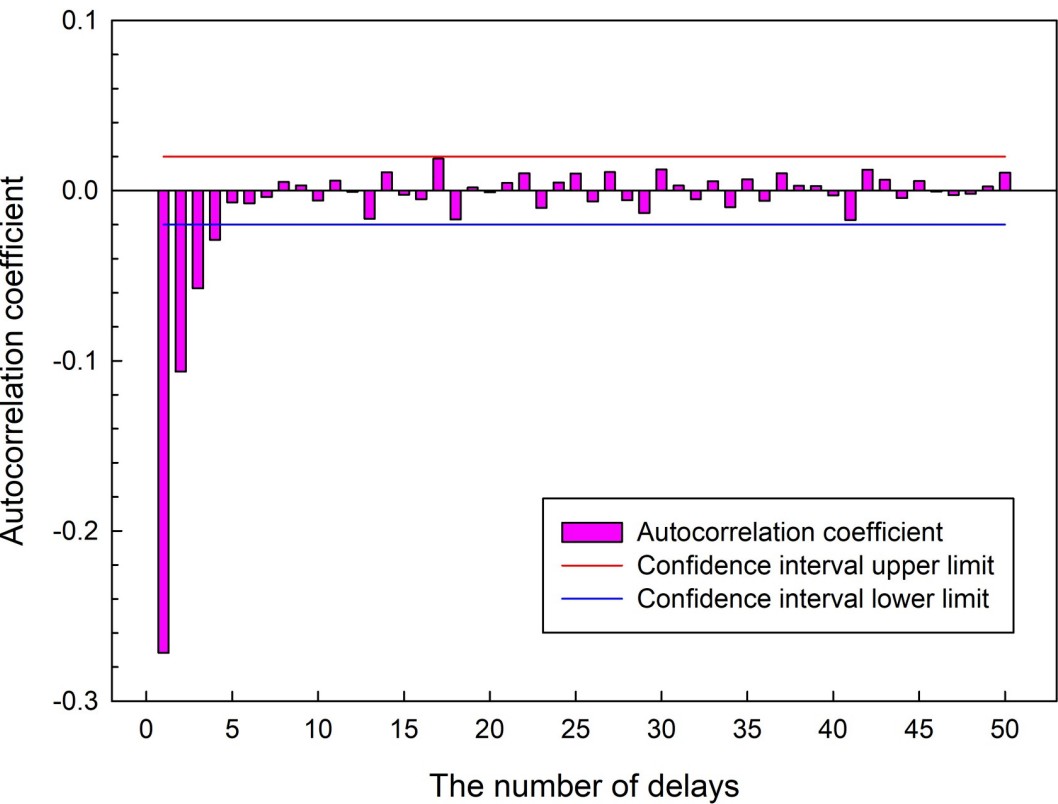

**Fig 5. The partial autocorrelation coefficient of daily ET$_o$.**

### Experimental and parameter setting

Daily ET$_o$ sequence data from the study site were recorded from 1980 to 2019, over the 40-year period in question, there have been marked changes in global ambient temperature and other weather data. In this study, the ET$_o$ was calculated by mixing six weather parameters, it can be seen from Table 1 that the standard deviation of the ET$_o$ is much smaller than the standard deviation of five weather parameters at each station. The research interval in this paper is 1 day, and the change of meteorological parameters in one day is random, this paper cannot accurately describe the change of meteorological parameters, so this paper ignores the influence of meteorological parameter changes on the prediction model. The data were split into training set (1980–2007), test set (2008–2018) and prediction set (2019), the data number of training set, test set and prediction set were 10227, 4021, and 365, respectively. Since the ET$_o$ are all positive, the sequence data of ET$_o$ was normalized using the function of MinMaxScaler in Pandas (https://pandas.pydata.org/) [33], and transform the data into the range [0, 1]. After training of models, the data were back transformed to the original scale.

Based on preliminary training, under 20 particles, the training time of the single hidden layer LSTM model is not less than 4h, For the four stations, the time cost of building the prediction models separately is huge and difficult. Given this, the prediction model was trained using data from the WG station. According to relevant literature [34], the setting of the hyperparameters for the hybrid model adopts the grid search method and the swarm intelligence method. In this study, the first model was used for preliminary experiments with

hyperparameter setting, it was determined that the $N_1$ is from 10 to 60, the $T_1$ is from 1 to 50, the activation function = ReLU, optimizer = Adam, dropout = 0.1, batch size = 64 and the number of train steps is 500. At the same time, the PSO algorithm parameters are set as: w = 0.5, $c_1 = c_2 = 0.5$, and the number of particles = 20.

After the first model experiment, in order to obtain better performing model, we increased the range of hidden layers and hyperparameters of the model, and the second model was constructed. It was determined that the $N_2$ is from 1 to 200, the $N_3$ is from 1 to 200, the $T_2$ is from 1 to 200, dropout = 0.1,0.2, or 0.3, the activation function = ReLU, optimizer = Adam, batch size = 64, and the number of train steps is 100. At the same time, to ensure a fair comparison, the PSO algorithm parameters are set as: w = 0.5, $c_1 = c_2 = 0.5$, and the number of particles = 20.

This paper uses the Keras framework (https://keras.io/) [35], which is an API designed for human beings, not machines, and regards the popular Deep Learning framework TensorFlow as the backstage supporter, to build the prediction model. The CPU in the experimental is Inter(R) Core (TM) i5 8500 @3.00GHz, and the RAM is 8GB, the version of Python is 3.6, and the software platform is the PyCharm 2021.3.

In order to evaluate the performance of the optimized model at other three stations, the optimized model was evaluated using the data from other three stations. At the same time, in order to verify the accuracy of the optimized model under different period, four types of periods are divided, including type A (training set 1980–2007,10220, test set 2008–2018, 4015), type B (training set 1980–2017,13869, test set, 1981–2018,13869), type C (training set 2003–2014, 4380, test set, 2015–2018,1460) and type D (training set 2011–2017, 2190, test set, 2017–2018, 730). The amount of data for each type of training set and test set is shown in the (Fig 6). The optimized model is trained again using the above training set, and the following evaluation criteria are used to evaluate hybrid model.

## Model evaluation criteria

Four evaluation measures were selected to indicate the performance of the different models [36].

Mean Absolute Error (MAE) is:

$$MAE = \frac{1}{N} \sum_{i=1}^{N} |x_i - y_i| \tag{12}$$

The Mean Squared Error (MSE) is:

$$MSE = \frac{1}{N} \sum_{i=1}^{N} (x_i - y_i)^2 \tag{13}$$

The root mean square error (RMSE) is:

$$RMSE = \sqrt{\frac{1}{N} \sum_{i=1}^{N} (x_i - y_i)^2} \tag{14}$$

R Squared ($R^2$) is:

$$R^2 = 1 - \frac{\sum_i (\hat{y}_i - y_i)}{\sum_i (\bar{y}_i - y_i)} \tag{15}$$

In the above formula, $y_i$ represents the predicted value. $x_i$ represents the true value. $\bar{y}_i$ is the average value. N represent the number of prediction value. MAE is the mean absolute error. It can reflect the actual situation of the predicted value error. The MSE is the expected value of

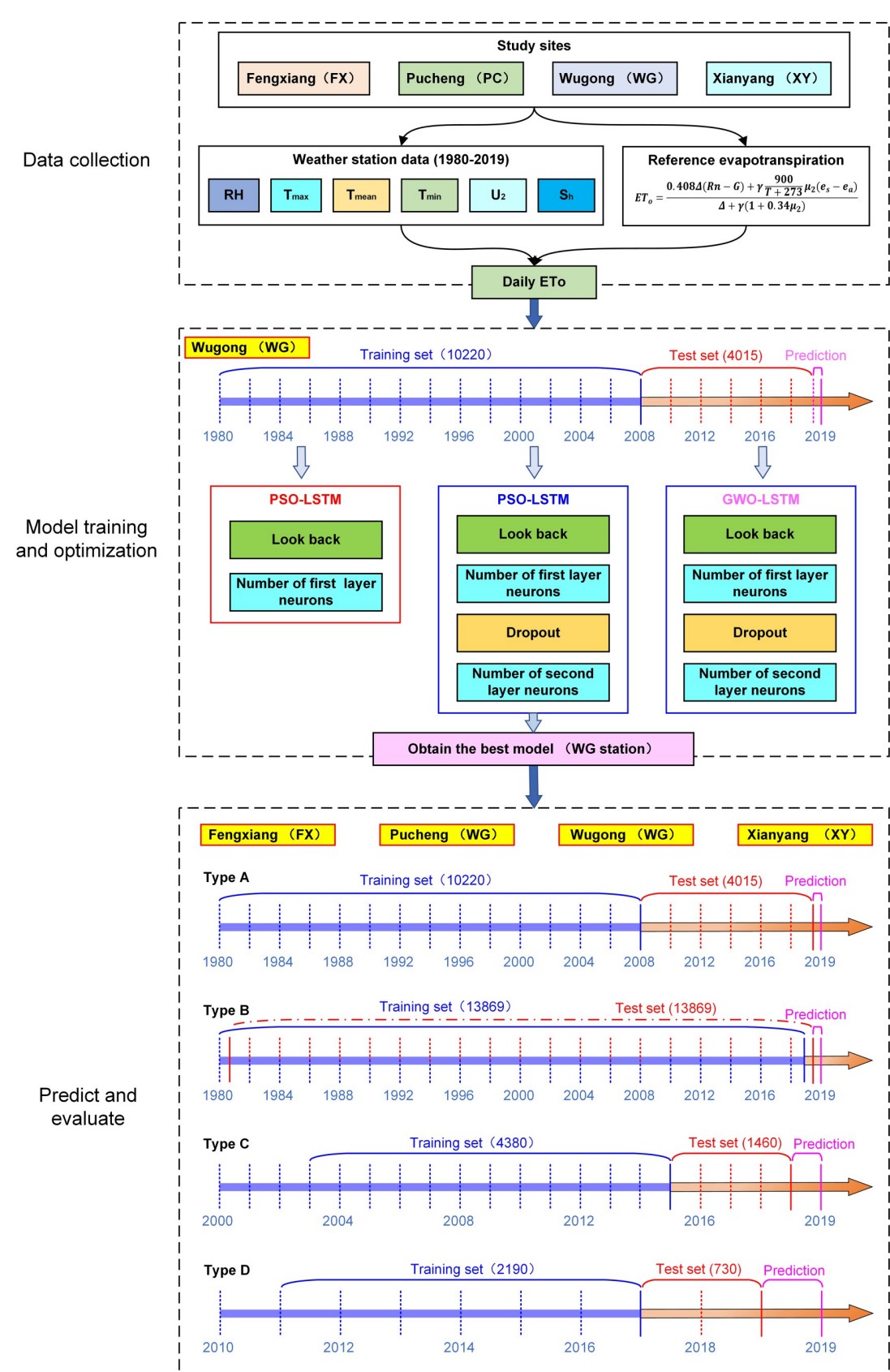

**Fig 6. The framework diagram of the research process.**

the square of the difference between the parameter estimate and the parameter true value, it can evaluate the degree of the data change, and the smaller value of the MSE, the better accuracy of the prediction model. RMSE is the square root of MSE. $R^2$ can eliminate the influence of dimension on evaluation measure.

# Results and discussion

## The result of the first optimization model

For the first optimized model, the PSO-LSTM optimization iterated 20 times. (Fig 7) present the training time, maximum fitness, average fitness and minimum fitness in each iteration. From the figures, we can find that the training time cost increases rapidly with the number of iterations. Although the batch size = 64, $T_1$ and $N_1$ of hybrid model is [1,50] and [10, 60], respectively, the utilization of CPU gradually decreases during the training process. The training time for the first and last iterations is 4.27h and 17.30h, respectively, and the total training time of 20 times iterations is 221.25h (9.21 days). The fitness gradually decreases as the number of iterations increases, and the minimum fitness is much smaller than the maximum fitness of each iteration. After 20 times iterations, the maximum, average and minimum fitness of optimal particle are 0.637, 0.608 and 0.597, respectively. After 20 times iterations, the hyperparameter combination corresponding to the minimum fitness is [5, 24], it means that the look back = 5, the number of first layer neurons = 24 is the optimal combination for the first model.

**Prediction and evaluation of the first model.** In order to evaluate the performance of model, the optimized model was trained again using the four different data set (type A, type B, type C and type D) for each station. The number of training steps during the optimization process is 500, in order to analyze the effect of different training steps on the model accuracy, the

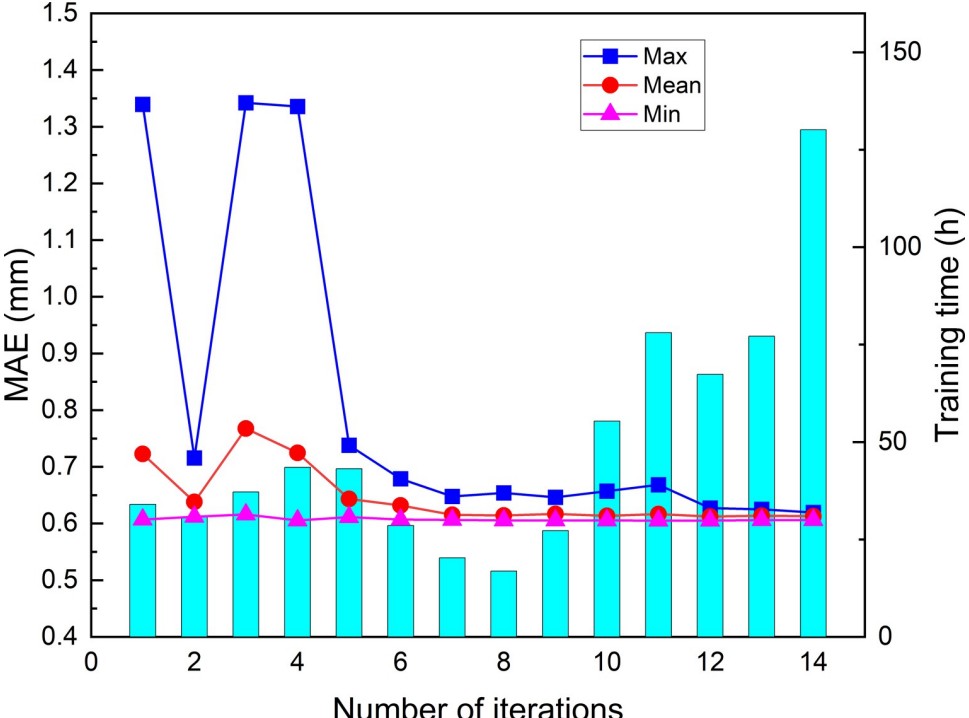

**Fig 7. The training time and fitness of the first model for each iteration.**

**Table 2. Comparison of the first model training result under different data set.**

| Site | Train steps | Type A | | | | Type B | | | |
|---|---|---|---|---|---|---|---|---|---|
| | | Training set | | Test set | | Training set | | Test set | |
| | | RMSE | MAE | RMSE | MAE | RMSE | MAE | RMSE | MAE |
| WG | 500 | 0.818 | 0.572 | 0.852 | 0.599 | 0.822 | 0.575 | 0.821 | 0.574 |
| | 1000 | 0.809 | 0.562 | 0.858 | 0.600 | 0.812 | 0.565 | 0.812 | 0.565 |
| | 1500 | 0.808 | 0.564 | 0.861 | 0.606 | 0.809 | 0.563 | 0.809 | 0.563 |
| | 2000 | 0.804 | 0.560 | 0.858 | 0.605 | 0.817 | 0.570 | 0.817 | 0.569 |
| FX | 500 | 0.848 | 0.609 | 0.928 | 0.675 | 0.871 | 0.624 | 0.872 | 0.625 |
| | 1000 | 0.836 | 0.601 | 0.929 | 0.680 | 0.875 | 0.629 | 0.875 | 0.625 |
| | 1500 | 0.839 | 0.595 | 0.948 | 0.688 | 0.847 | 0.603 | 0.849 | 0.605 |
| | 2000 | 0.831 | 0.589 | 0.950 | 0.691 | 0.852 | 0.607 | 0.854 | 0.609 |
| XY | 500 | 0.919 | 0.648 | 0.941 | 0.664 | 0.916 | 0.644 | 0.917 | 0.644 |
| | 1000 | 0.911 | 0.643 | 0.934 | 0.665 | 0.918 | 0.651 | 0.919 | 0.651 |
| | 1500 | 0.902 | 0.626 | 0.959 | 0.674 | 0.918 | 0.636 | 0.919 | 0.637 |
| | 2000 | 0.896 | 0.623 | 0.955 | 0.671 | 0.901 | 0.635 | 0.902 | 0.635 |
| PC | 500 | 0.946 | 0.678 | 0.930 | 0.673 | 0.957 | 0.675 | 0.955 | 0.673 |
| | 1000 | 0.941 | 0.665 | 0.943 | 0.674 | 0.948 | 0.666 | 0.946 | 0.665 |
| | 1500 | 0.943 | 0.668 | 0.943 | 0.673 | 0.939 | 0.658 | 0.937 | 0.656 |
| | 2000 | 0.924 | 0.653 | 0.938 | 0.673 | 0.913 | 0.642 | 0.913 | 0.642 |
| Site | Train steps | Type C | | | | Type D | | | |
| | | Training set | | Test set | | Training set | | Test set | |
| | | RMSE | MAE | RMSE | MAE | RMSE | MAE | RMSE | MAE |
| WG | 500 | 0.813 | 0.555 | 0.901 | 0.649 | 0.827 | 0.577 | 0.902 | 0.651 |
| | 1000 | 0.812 | 0.553 | 0.909 | 0.655 | 0.810 | 0.567 | 0.914 | 0.663 |
| | 1500 | 0.800 | 0.550 | 0.912 | 0.658 | 0.808 | 0.652 | 0.911 | 0.653 |
| | 2000 | 0.802 | 0.553 | 0.907 | 0.658 | 0.785 | 0.535 | 0.949 | 0.685 |
| FX | 500 | 0.889 | 0.633 | 0.948 | 0.686 | 0.943 | 0.676 | 0.941 | 0.687 |
| | 1000 | 0.877 | 0.616 | 0.966 | 0.699 | 0.907 | 0.641 | 0.952 | 0.688 |
| | 1500 | 0.887 | 0.627 | 0.959 | 0.694 | 0.915 | 0.646 | 0.978 | 0.699 |
| | 2000 | 0.874 | 0.623 | 0.966 | 0.701 | 0.900 | 0.641 | 0.949 | 0.689 |
| XY | 500 | 0.903 | 0.626 | 0.979 | 0.693 | 0.938 | 0.651 | 1.018 | 0.720 |
| | 1000 | 0.895 | 0.626 | 0.981 | 0.703 | 0.925 | 0.653 | 1.006 | 0.725 |
| | 1500 | 0.898 | 0.620 | 0.988 | 0.700 | 0.857 | 0.601 | 1.063 | 0.755 |
| | 2000 | 0.882 | 0.603 | 0.997 | 0.702 | 0.904 | 0.635 | 1.038 | 0.739 |
| PC | 500 | 0.925 | 0.662 | 0.911 | 0.655 | 0.922 | 0.660 | 0.919 | 0.654 |
| | 1000 | 0.921 | 0.641 | 0.927 | 0.656 | 0.912 | 0.640 | 0.944 | 0.663 |
| | 1500 | 0.912 | 0.644 | 0.916 | 0.657 | 0.909 | 0.634 | 0.940 | 0.655 |
| | 2000 | 0.908 | 0.634 | 0.920 | 0.653 | 0.902 | 0.638 | 0.948 | 0.671 |

number of training steps were set as 500, 1000, 1500, and 2000, respectively. Table 2 shows the training results.

For the four stations and four datasets, it can be seen from the table that the most RMSE and MAE of training set gets smaller with the number of training steps increases, on the contrary, the RMSE and MAE of test set get bigger with the number of training steps increases. Therefore, when the number of training steps is 500, the RMSE and MAE of test set are the smallest among the four training steps. The model trained when the number of training steps is 500 can be better used to predict the $ET_o$.

**Table 3. The forecast results of first optimized model under different data set.**

| Site | Type A | | | | Type B | | | |
|------|--------|--------|--------|--------|--------|--------|--------|--------|
| | MSE | RMSE | MAE | $R^2$ | MSE | RMSE | MAE | $R^2$ |
| WG | 0.895 | 0.946 | 0.661 | 0.814 | 0.891 | 0.944 | 0.660 | 0.816 |
| FX | 0.905 | 0.951 | 0.687 | 0.823 | 0.935 | 0.967 | 0.697 | 0.817 |
| XY | 0.961 | 0.980 | 0.677 | 0.804 | 0.978 | 0.989 | 0.686 | 0.801 |
| PC | 0.878 | 0.937 | 0.676 | 0.858 | 0.903 | 0.950 | 0.664 | 0.860 |
| Site | Type C | | | | Type D | | | |
| | MSE | RMSE | MAE | $R^2$ | MSE | RMSE | MAE | $R^2$ |
| WG | 0.935 | 0.967 | 0.678 | 0.806 | 0.956 | 0.978 | 0.679 | 0.804 |
| FX | 0.906 | 0.952 | 0.684 | 0.824 | 0.932 | 0.966 | 0.693 | 0.819 |
| XY | 0.891 | 0.944 | 0.662 | 0.819 | 1.034 | 1.017 | 0.694 | 0.806 |
| PC | 0.912 | 0.955 | 0.684 | 0.852 | 0.895 | 0.946 | 0.673 | 0.855 |

The first optimized model was used to predict $ET_o$ data in 2019, and the forecast results are shown in Table 3. As we can see from the table the MSE, RMSE, MAE and $R^2$ between the actual value and predicted value of four different data set are similar for each site, which shows that the optimized model is suitable for these four different datasets. Based on the MAE, the data types with the highest prediction accuracy for the four stations are Type B, Type C, Type C, and Type B, respectively.

The prediction results with the highest prediction accuracy are shown in (Fig 8). As we can see from the figure that the change trend of between the predicted value and the actual value is similar, and the predicted value of the optimized model is relatively close to the true value at each station. The model predicted $ET_o$ value is less than the actual value from March to September, and the model predicted and actual $ET_o$ value are very similar from the October to the February, and the four stations have the same result.

Statistics of the error distribution between the predicted value of the model and the actual value in 2019 are shown in (Fig 9). From the figure, the errors of model predictions for the four stations are all within [−4,4] mm, and the range of the most of the prediction residual data was -1.0 to 1.0 mm. For four stations, the number of predicted values higher than the true value was 183, 164, 148, and 199, respectively, where the minimum error was -3.474 mm, -3.429 mm, -3.369 mm, and -2.239 mm, respectively. and the maximum error was 3.279 mm, 3.470 mm, 3.313 mm, and 4.155 mm, respectively. The main reason for this result is that due to the random fluctuation of meteorological data, it is difficult for the hybrid model to form a high-precision simulation of stochastic fluctuations.

In order to find out the correlation between prediction and true value, the fitting between the prediction and true values for validation set was computed. The fitting between the predicted and actual values for four stations in 2019 is shown in (Fig 10). For the four stations, the correlation coefficient of the optimized model for test set is 0.816, 0.824, 0.819, and 0.860, respectively. The experiments above also show that the first optimized model displays good prediction accuracy. It remains within a stable acceptable error range, which ensures that the daily $ET_o$ data predicted by the model can be used in actual guidance.

**The result of the second optimization model.** For the second optimized model, the PSO-LSTM optimization iterated 9 times. (Fig 11) present the training time, maximum, average and minimum fitness in each iteration. From the figures, we can find that the training time increases rapidly with the number of iterations. Although the batch size = 64, the $T_2$, $N_2$,

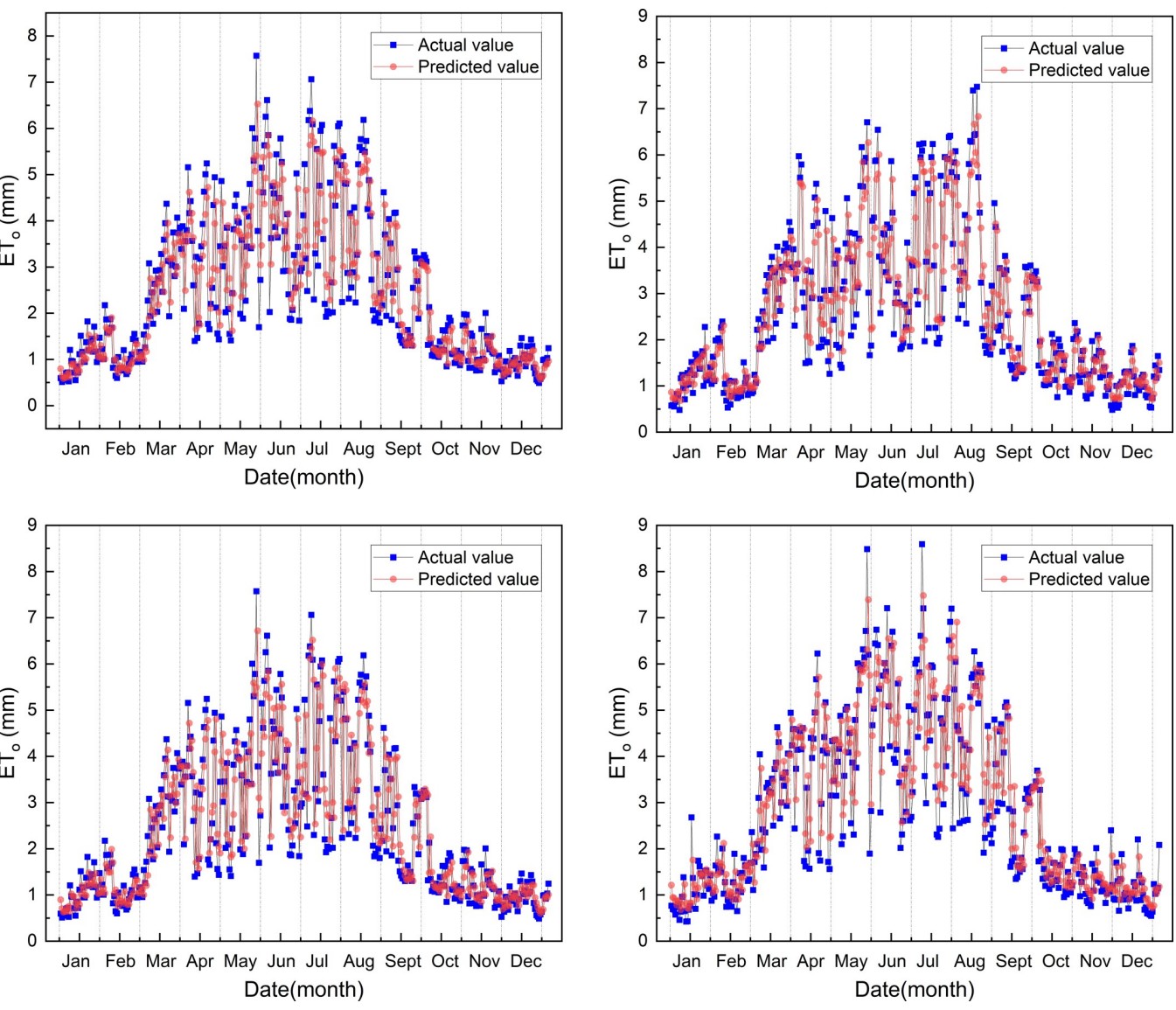

**Fig 8. Comparison of predicted and actual ET$_o$ for the first optimized model in 2019.**

and N$_3$ of hybrid model is [1,200], [1,200], and [1, 200], respectively, and D = 0.1,0.2, and 0.3, the utilization of CPU gradually decreases during the training process. The training time for the first and last iterations is 18.20h and 147.84h, respectively, and the total training time of 9 times iterations is 806.03h (33.58 days). The fitness gradually decreases as the number of iterations increases, and the minimum fitness is much smaller than the maximum fitness of each iteration. After 9 times iterations, the maximum, average and minimum are 0.619, 0.606 and 0.600, respectively. After 9 times iterations, the hyperparameter combination of the optimized model corresponding to the minimum fitness is [22, 175, 39, 0.2], it means that the look back = 22, the number of first layer neurons = 175, the number of second layer neurons = 39, and dropout = 0.2 is the optimal combination for the model.

**Prediction and evaluation of the second model.** In order to evaluate the performance of the second optimized model, the model was also trained again using the four different data set

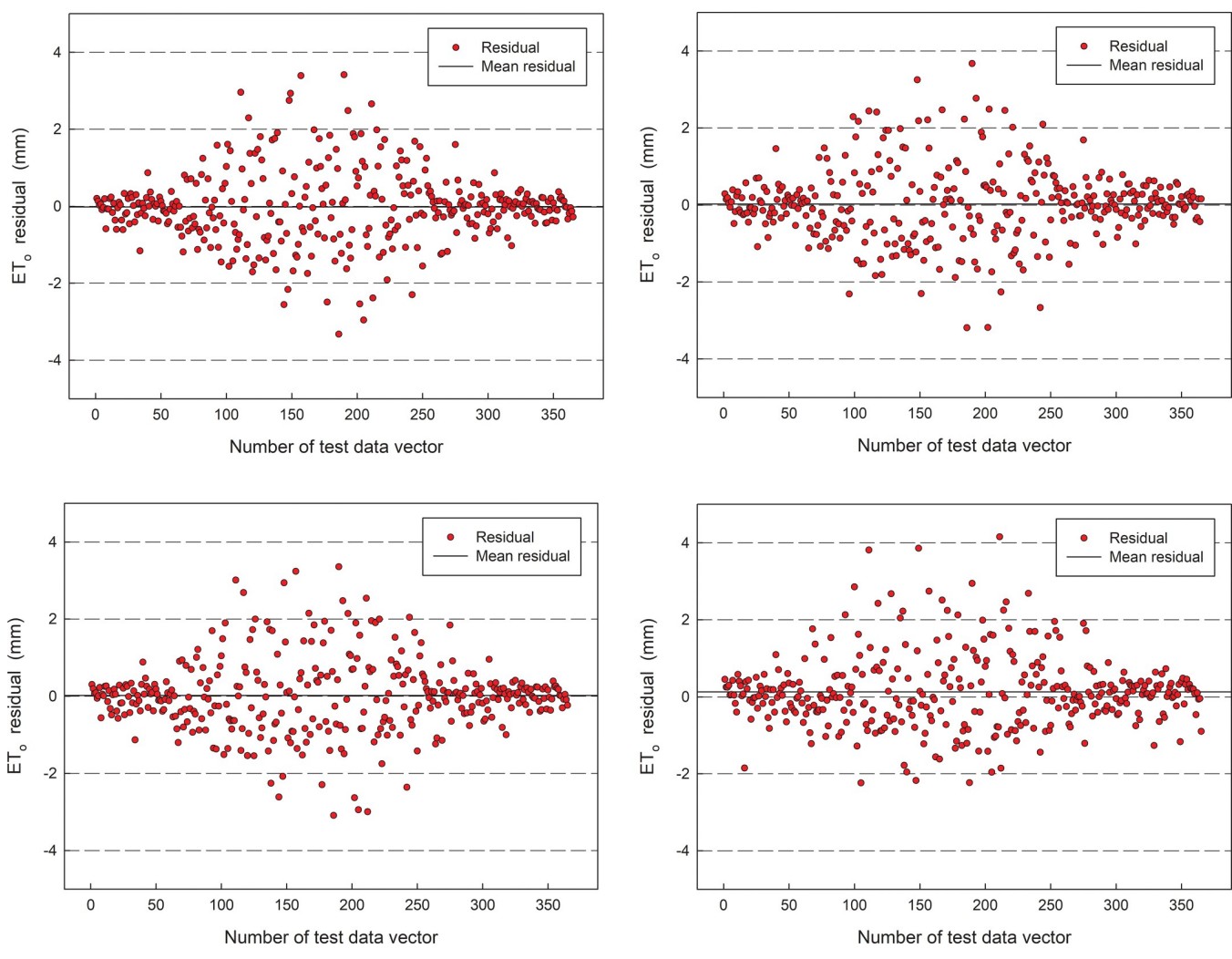

**Fig 9. The residual of prediction $ET_o$ for the first optimized model in 2019.**

of each station. The number of training steps during the optimization process is 100, in order to study the effect of different training steps on the model accuracy, the number of training steps were set as 50, 100, 150 and 200, respectively. Table 4 shows the training results.

Similar to the result of the first model, for each station and each dataset, it can be seen from the table that the RMSE and MAE of most training set gets smaller with the number of training steps increases, on the contrary, the RMSE and MAE of most test set get bigger with the number of training steps increases. Therefore, when the number of training steps is 50, the RMSE and MAE of test set are the smallest among the four training steps. Therefore, the model trained when the number of training steps is 50 can be better used to predict the $ET_o$ for each station.

The optimized model was also used to predict $ET_o$ data in 2019, and the forecast results are shown in Table 5. As we can see from the table the MSE, RMSE, MAE and $R^2$ between the actual value and predicted value under four different data set are similar, which shows that the optimized model is suitable for these four different datasets. Based on the MAE, the data types

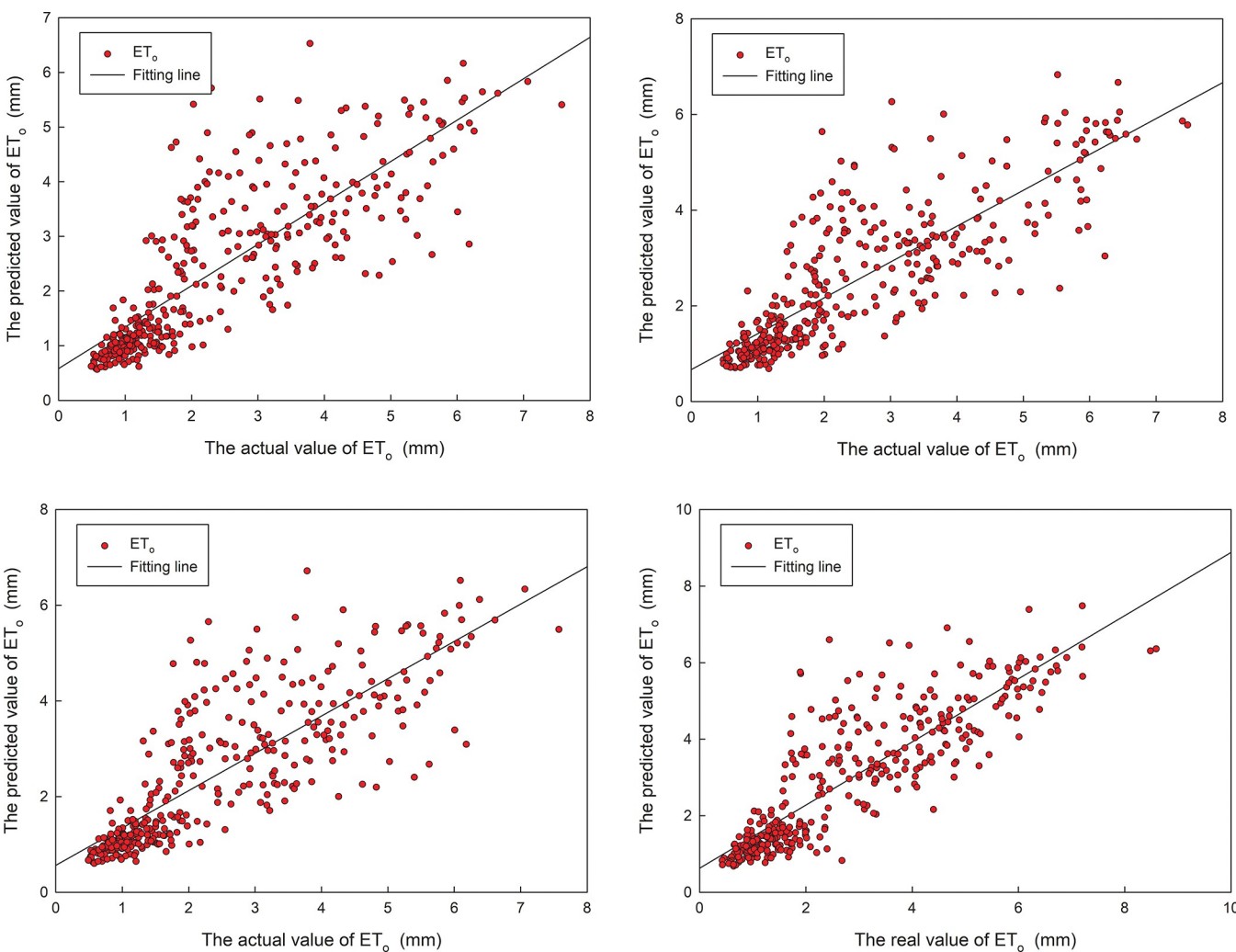

**Fig 10. The fitting between predicted and actual values for the first optimized model in 2019.**

with the highest prediction accuracy for the four stations are Type C, Type C, Type D, and Type B, respectively.

Similar to the result of the first model, the prediction results with the highest prediction accuracy are shown in (Fig 12). As we can see from the figure that the change trend of between the predicted value and the actual value is similar, and the prediction value of the optimized model is relatively close to the true value at each station. The predicted $ET_o$ value of the second model is also less than the actual value from March to September, and the predicted $ET_o$ value and actual value are also very close to the value from the October to the February. All four stations have the same result.

Statistics of the error distribution between the predicted value of the model and the actual value in 2019 are shown in (Fig 13). From the figure, the errors of model predictions for the four stations are also all within [−4,4] mm, and the range of the most of the prediction residual data was -1.0 to 1.0 mm. For four stations, the number of predicted values higher than the true value was 179, 189, 191, and 187, respectively. where the minimum error was -3.323mm, -3.192mm, -3.091mm, and -2.712mm, respectively. and the maximum error was 3.4149 mm,

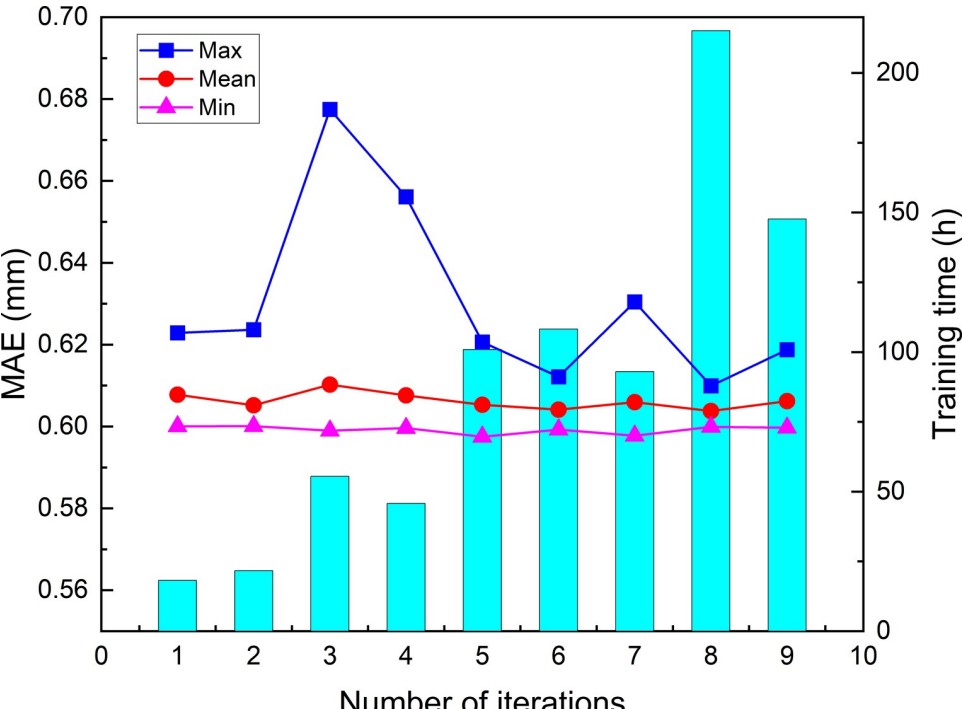

**Fig 11. The training time and fitness of the second model for each iteration.**

3.671mm, 3.357mm, and 3.748mm, respectively. The main reason for this result also is that due to the random fluctuation of meteorological data, it is difficult for the hybrid model to form a high-precision simulation of stochastic fluctuations.

The fitting between the prediction value and actual values in 2019 is shown in (Fig 14). For the four stations, the correlation coefficient of the optimized model for test set is 0.818, 0.823, 0.822, and 0.857, respectively. The experiments above also show that the first optimized model displays good prediction accuracy. It remains within a stable acceptable error range, which ensures that the daily $ET_o$ data predicted by the model can be used in actual guidance.

**Comparison of two models.** It can be seen from the two PSO-LSTM predicted results, the January, August, November and December were the months with the higher proportion of absolute errors, mainly because the value of daily $ET_o$ in these months is small, and smaller than the value of daily $ET_o$ in other months. The February, March, April and May are the months with the lower proportion of absolute error. Although the errors are relatively large in June, July, October and September, the overall errors are controlled within the range of from –1.0 to 1.0 mm. At the same time, June, July, October and September are not critical periods for the growth of crops, so the relatively low prediction accuracy has no impact on crop irrigation decision.

Compared to the first model, the second model has a larger range of hyperparameters and a more complex model topology, even though the second model has a much smaller number of training steps (100) than the first model (500), the training time of the second model is much greater than the training time cost of the first model. In fact, in the iterative process, the changes of fitness values of the two hybrid models are not significant. There are many factors that affect the performance of the hybrid model, including the characteristics of data, the hyperparameters in model and training methods. During the optimization process, the two

**Table 4. Comparison of the second model training result under different data set.**

| Site | Train steps | Type A | | | | Type B | | | |
|---|---|---|---|---|---|---|---|---|---|
| | | Training set | | Test set | | Training set | | Test set | |
| | | RMSE | MAE | RMSE | MAE | RMSE | MAE | RMSE | MAE |
| WG | 50 | 0.827 | 0.582 | 0.859 | 0.607 | 0.829 | 0.579 | 0.829 | 0.578 |
| | 100 | 0.816 | 0.567 | 0.858 | 0.602 | 0.818 | 0.571 | 0.817 | 0.571 |
| | 150 | 0.799 | 0.551 | 0.870 | 0.606 | 0.816 | 0.559 | 0.817 | 0.560 |
| | 200 | 0.785 | 0.540 | 0.894 | 0.622 | 0.776 | 0.542 | 0.777 | 0.543 |
| FX | 50 | 0.855 | 0.611 | 0.931 | 0.673 | 0.870 | 0.627 | 0.871 | 0.627 |
| | 100 | 0.859 | 0.611 | 0.936 | 0.676 | 0.881 | 0.637 | 0.882 | 0.637 |
| | 150 | 0.841 | 0.602 | 0.936 | 0.682 | 0.863 | 0.617 | 0.864 | 0.618 |
| | 200 | 0.848 | 0.592 | 0.973 | 0.690 | 0.857 | 0.613 | 0.858 | 0.613 |
| XY | 50 | 0.905 | 0.646 | 0.920 | 0.662 | 0.908 | 0.651 | 0.909 | 0.651 |
| | 100 | 0.914 | 0.637 | 0.943 | 0.660 | 0.919 | 0.641 | 0.919 | 0.641 |
| | 150 | 0.893 | 0.641 | 0.921 | 0.668 | 0.914 | 0.633 | 0.915 | 0.634 |
| | 200 | 0.877 | 0.617 | 0.941 | 0.670 | 0.879 | 0.611 | 0.882 | 0.613 |
| PC | 50 | 0.906 | 0.646 | 0.920 | 0.662 | 0.908 | 0.651 | 0.909 | 0.651 |
| | 100 | 0.914 | 0.637 | 0.943 | 0.660 | 0.919 | 0.641 | 0.919 | 0.641 |
| | 150 | 0.893 | 0.641 | 0.921 | 0.668 | 0.914 | 0.633 | 0.915 | 0.634 |
| | 200 | 0.877 | 0.617 | 0.941 | 0.671 | 0.879 | 0.611 | 0.882 | 0.613 |

| Site | Train steps | Type C | | | | Type D | | | |
|---|---|---|---|---|---|---|---|---|---|
| | | Training set | | Test set | | Training set | | Test set | |
| | | RMSE | MAE | RMSE | MAE | RMSE | MAE | RMSE | MAE |
| WG | 50 | 0.820 | 0.566 | 0.895 | 0.648 | 0.836 | 0.597 | 0.899 | 0.664 |
| | 100 | 0.819 | 0.566 | 0.905 | 0.656 | 0.845 | 0.604 | 0.905 | 0.662 |
| | 150 | 0.802 | 0.559 | 0.892 | 0.651 | 0.836 | 0.591 | 0.917 | 0.671 |
| | 200 | 0.799 | 0.557 | 0.923 | 0.674 | 0.818 | 0.581 | 0.923 | 0.677 |
| FX | 50 | 0.907 | 0.649 | 0.961 | 0.707 | 0.950 | 0.678 | 0.955 | 0.700 |
| | 100 | 0.915 | 0.644 | 0.969 | 0.691 | 0.932 | 0.677 | 0.942 | 0.695 |
| | 150 | 0.911 | 0.646 | 0.968 | 0.693 | 0.932 | 0.670 | 0.945 | 0.690 |
| | 200 | 0.886 | 0.633 | 0.953 | 0.695 | 0.921 | 0.675 | 0.941 | 0.695 |
| XY | 50 | 0.903 | 0.633 | 0.967 | 0.687 | 0.941 | 0.682 | 1.001 | 0.741 |
| | 100 | 0.902 | 0.629 | 0.979 | 0.699 | 0.919 | 0.664 | 0.989 | 0.729 |
| | 150 | 0.889 | 0.618 | 0.977 | 0.693 | 0.920 | 0.653 | 0.997 | 0.726 |
| | 200 | 0.880 | 0.602 | 0.995 | 0.698 | 0.928 | 0.672 | 1.023 | 0.758 |
| PC | 50 | 0.903 | 0.633 | 0.967 | 0.687 | 0.941 | 0.682 | 1.001 | 0.741 |
| | 100 | 0.902 | 0.629 | 0.979 | 0.699 | 0.919 | 0.664 | 0.989 | 0.729 |
| | 150 | 0.889 | 0.618 | 0.977 | 0.694 | 0.920 | 0.653 | 0.997 | 0.726 |
| | 200 | 0.880 | 0.602 | 0.995 | 0.698 | 0.928 | 0.673 | 1.023 | 0.758 |

models obtained the prediction 400 and 180 results under different hyperparameters combinations, respectively. Due to the long iteration time, the hybrid model unable to training more iterations in a short time. For the four datasets, each station obtained 8 optimized prediction models, at the same time, the prediction accuracy of the 8 models is similar. For the four sites, this study can meet the forecasting needs under different dataset. However, the optimized hybrid models provided only slight performance gains in the study.

it is impossible to consider all factors in the study. The optimized models have not exhibited high accuracies, Furthermore, deep learning models generally have more hyperparameters to be optimized with the meteorological data features.

**Table 5. The forecast results of second optimized model under different data set.**

| Site | Type A | | | | Type B | | | |
|---|---|---|---|---|---|---|---|---|
| | MSE | RMSE | MAE | $R^2$ | MSE | RMSE | MAE | $R^2$ |
| WG | 0.870 | 0.933 | 0.665 | 0.817 | 0.896 | 0.947 | 0.673 | 0.816 |
| FX | 0.923 | 0.961 | 0.692 | 0.823 | 0.919 | 0.959 | 0.694 | 0.822 |
| XY | 0.947 | 0.973 | 0.686 | 0.804 | 0.940 | 0.969 | 0.689 | 0.805 |
| PC | 0.884 | 0.940 | 0.684 | 0.856 | 0.884 | 0.940 | 0.678 | 0.857 |
| Site | Type C | | | | Type D | | | |
| | MSE | RMSE | MAE | $R^2$ | MSE | RMSE | MAE | $R^2$ |
| WG | 0.870 | 0.933 | 0.661 | 0.818 | 0.866 | 0.930 | 0.666 | 0.820 |
| FX | 0.906 | 0.952 | 0.691 | 0.823 | 0.940 | 0.970 | 0.704 | 0.823 |
| XY | 0.965 | 0.982 | 0.679 | 0.806 | 0.866 | 0.931 | 0.674 | 0.822 |
| PC | 0.887 | 0.942 | 0.682 | 0.855 | 0.952 | 0.976 | 0.712 | 0.857 |

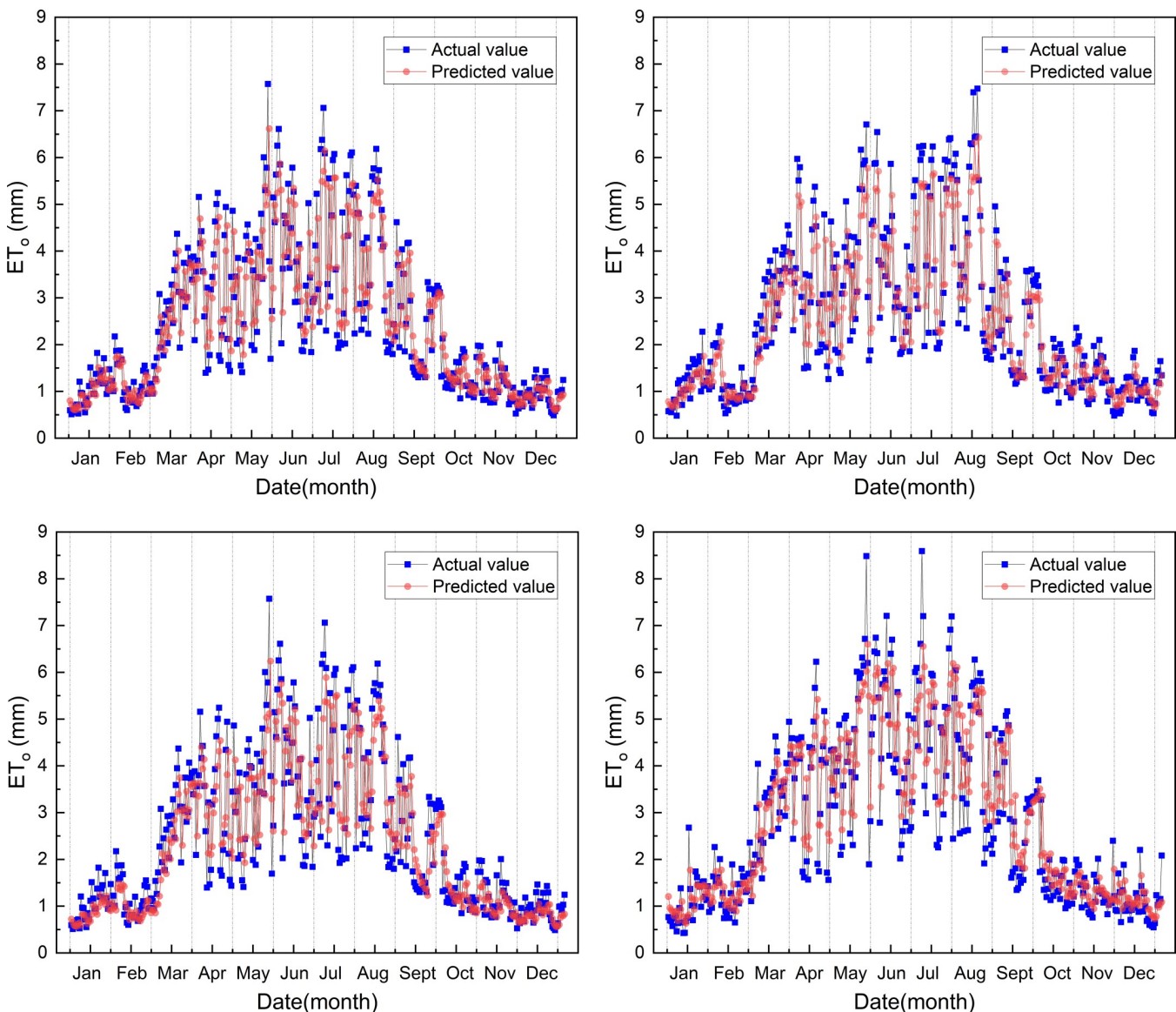

**Fig 12. Comparison of predicted and actual ET$_o$ for the second optimized model in 2019.**

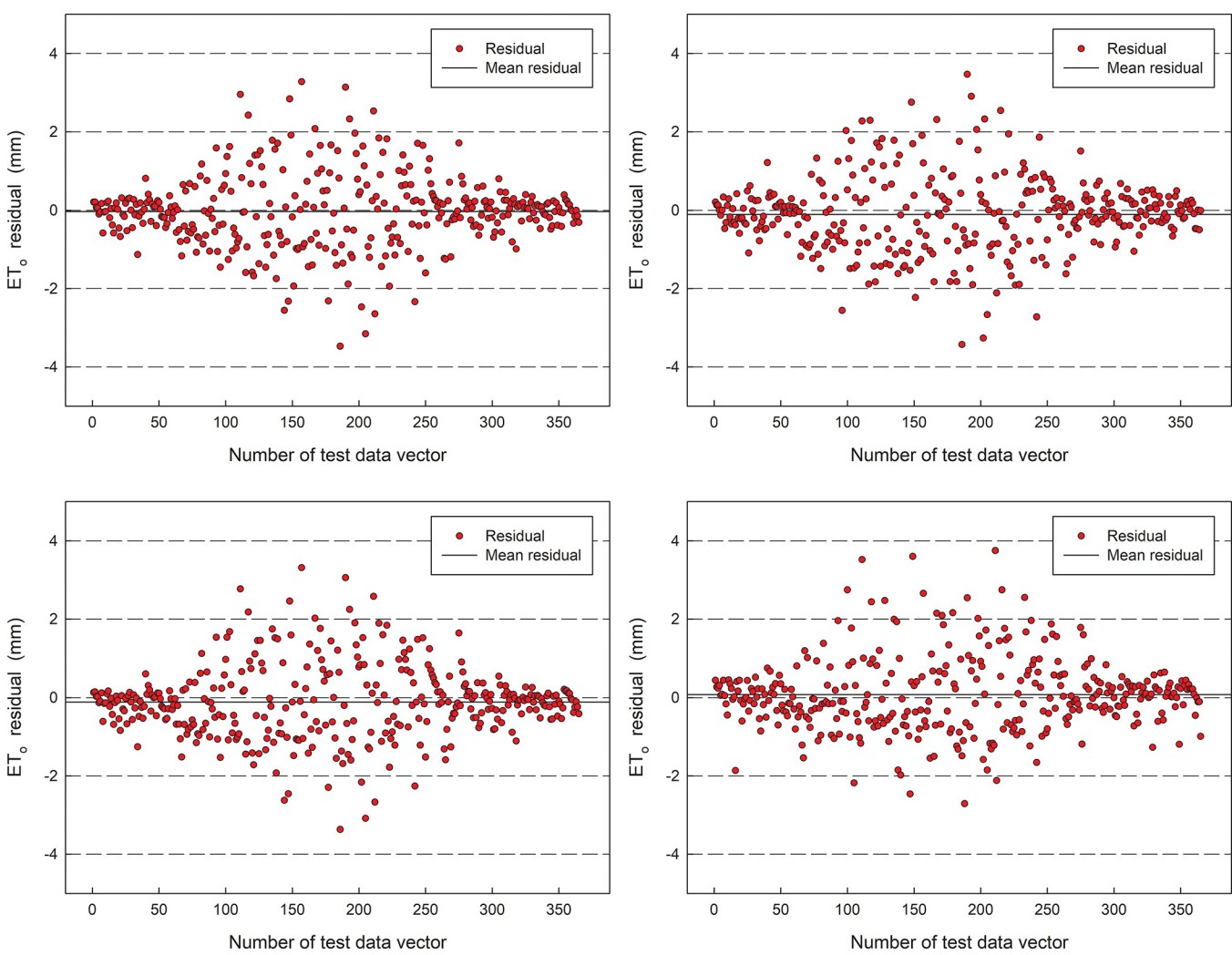

**Fig 13. The residual of prediction ETo for the second optimized model in 2019.**

## Conclusions

In this paper, we investigate the use of the LSTM neural network to predict daily $ET_o$. Two different topology LSTM models were constructed and optimized using the PSO algorithm hyperparameters in the LSTM neural network. The accuracy of the two hybrid models were evaluated using four different datasets in the WG, FX, XY, and PC stations, Shaanxi province, China. A single hidden layer LSTM model with 24 nodes was selected, and the value of look back was selected as 5. And the first and second hidden layer LSTM with 179 and 39, respectively, and the value of look back and dropout were selected as 22 and 0.2, respectively.

The optimized hybrid models were also predicted under different dataset, it can be found the optimized hybrid model has better accuracy under four different data set. Furthermore, the hybrid models developed in this study do not depend on external data, requiring only data measured at a local weather station, the successful validation of these models will allow agricultural producers to appropriately schedule the irrigation of crops according to $ET_o$ forecasts. In these situations, they can provide valuable information to improve tasks such as irrigation planning.

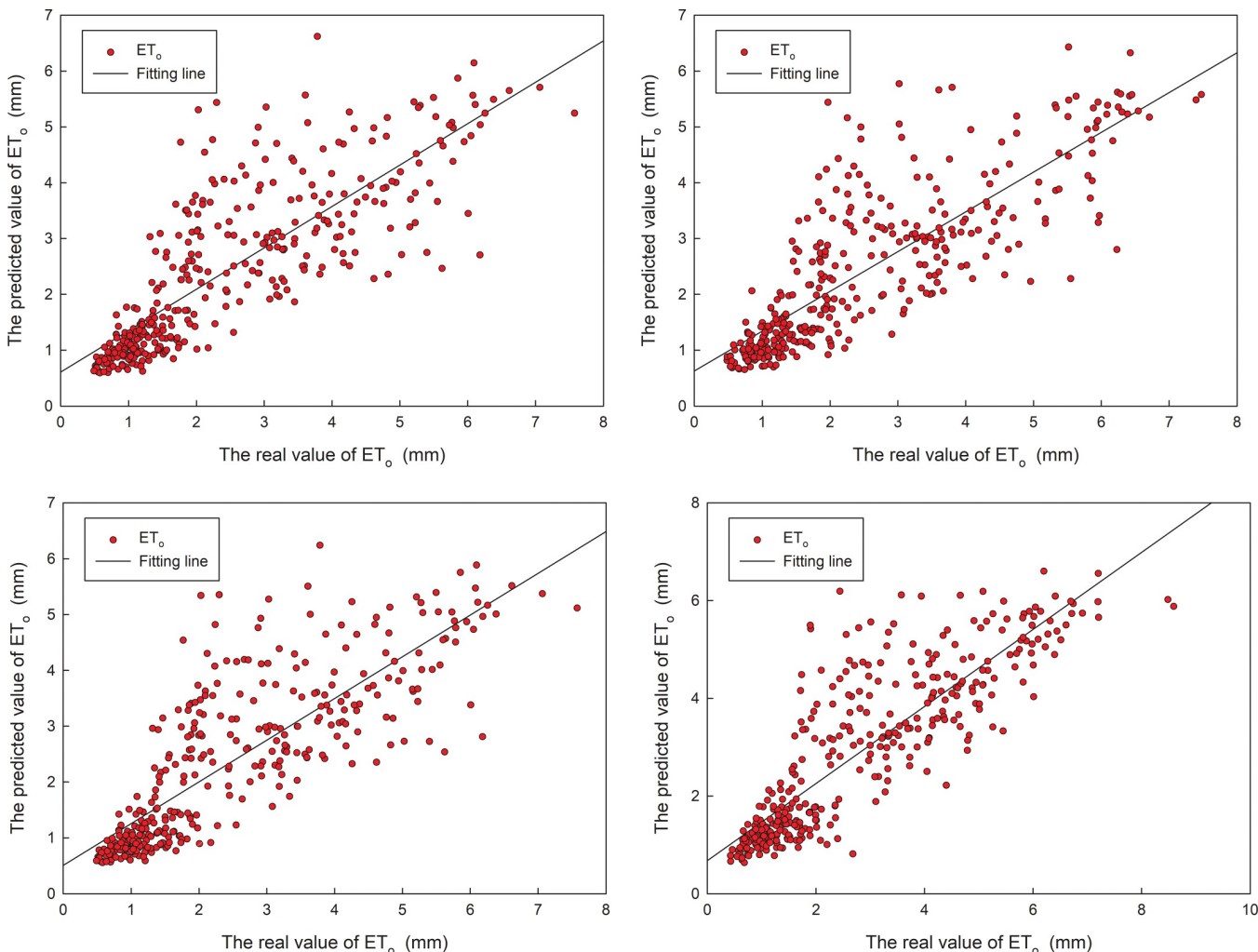

**Fig 14. The fitting between predicted and actual values for the second optimized model in 2019.**

## Supporting information

**S1 File. The weather data and $ET_o$ of the four stations.**
(XLSX)

**S2 File. The training time and fitness of two models for each iteration.**
(XLSX)

**S3 File. The result of training set and test set for the first model.**
(XLSX)

**S4 File. The result of prediction set for the first model.**
(XLSX)

**S5 File. The result of training set and test set for the second model.**
(XLSX)

**S6 File. The result of prediction set for the second model.**
(XLSX)

**S7 File. The results of grid search method for two models.**
(RAR)

## Author Contributions

**Data curation:** Weibing Jia, Zhenhao Zheng, Peijun Xie.

**Formal analysis:** Weibing Jia.

**Funding acquisition:** Zhengying Wei.

**Methodology:** Weibing Jia, Yubin Zhang.

**Resources:** Zhengying Wei, Zhenhao Zheng, Peijun Xie.

**Software:** Weibing Jia, Yubin Zhang.

**Supervision:** Zhengying Wei.

**Validation:** Weibing Jia, Yubin Zhang.

**Visualization:** Weibing Jia.

**Writing – original draft:** Weibing Jia, Yubin Zhang.

**Writing – review & editing:** Weibing Jia, Yubin Zhang, Zhengying Wei.

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
