## [Decision Letter · Decision Letter 0]

12 Jul 2022

PONE-D-22-09180Daily reference evapotranspiration prediction for irrigation scheduling decisions based on the hybrid PSO-LSTM modelPLOS ONE

Dear Dr. Jia,

Thank you for submitting your manuscript to PLOS ONE. After careful consideration, we feel that it has merit but does not fully meet PLOS ONE’s publication criteria as it currently stands. Therefore, we invite you to submit a revised version of the manuscript that addresses the points raised during the review process. I am indebted to the reviewers for providing extensive comments and suggestions. I recommend that you consider them carefully. In particular, two major issues must be addressed in your revision: the reasons for selecting PSO for the purposes described, and a more defensible comparison with other, contemporary methods. A number of other, more minor issues should be corrected, as noted in reviewer comments.

We look forward to receiving your revised manuscript.

Kind regards,

Andrew Lewis

Academic Editor

PLOS ONE

Journal Requirements:

2. Thank you for submitting the above manuscript to PLOS ONE. During our internal evaluation of the manuscript, we found significant text overlap between your submission and the following previously published works, some of which you are an author.

https://www.sciencedirect.com/science/article/pii/S1567422322000023?via%3Dihub

Please revise the manuscript to rephrase the duplicated text, cite your sources, and provide details as to how the current manuscript advances on previous work. Please note that further consideration is dependent on the submission of a manuscript that addresses these concerns about the overlap in text with published work.

4. We note that Figure 1 in your submission contain map images which may be copyrighted. All PLOS content is published under the Creative Commons Attribution License (CC BY 4.0), which means that the manuscript, images, and Supporting Information files will be freely available online, and any third party is permitted to access, download, copy, distribute, and use these materials in any way, even commercially, with proper attribution. For these reasons, we cannot publish previously copyrighted maps or satellite images created using proprietary data, such as Google software (Google Maps, Street View, and Earth). For more information, see our copyright guidelines: http://journals.plos.org/plosone/s/licenses-and-copyright.

a. You may seek permission from the original copyright holder of Figure(s) [#] to publish the content specifically under the CC BY 4.0 license. 

5. We note you have included a table to which you do not refer in the text of your manuscript. Please ensure that you refer to Table 9 in your text; if accepted, production will need this reference to link the reader to the Table.

Reviewers' comments:

Reviewer's Responses to Questions

**Comments to the Author**

1. Is the manuscript technically sound, and do the data support the conclusions?

Reviewer #1: Partly

Reviewer #2: No

2. Has the statistical analysis been performed appropriately and rigorously? 

Reviewer #1: N/A

Reviewer #2: No

3. Have the authors made all data underlying the findings in their manuscript fully available?

Reviewer #1: No

Reviewer #2: Yes

4. Is the manuscript presented in an intelligible fashion and written in standard English?

Reviewer #1: Yes

Reviewer #2: Yes

5. Review Comments to the Author

Reviewer #1: This paper presents a hybrid approach for predicting evapotranspiration levels. While the approach (producing a hybrid solver) is not particularly novel, the results themselves have some value. There are a number of ways in which this manuscript can be improved. These are detailed below and categorised as minor and major changes.

Minor:

* Abstract: This is far too long and in places has far too much detail. Lines 14-21 (manuscript line numbers) have far too much methodological detail and need to be summarised into just a couple of sentences.

* There are a few English expression issues through the manuscript. It needs to be proofed before the next revision.

* You need to use consistent math typesetting throughout the manuscript. One of the biggest issues is that variables are inconsistently italicised. Italicised variables tell the reader that an item is a variable. For example, in lines 129-148 no italicisation is present, however, in lines 308-315 it is. You need to carefully check the entire manuscript to make sure your mathematical representations are uniform.

* Lines 105-111: You refer to Section numbers in these lines, however, each section does not have a number, only a name. I suggest you just name the sections in these lines. Also, it makes no sense at this point to talk about what is in Section 1, as we are now at the end of Section 1. Remove this.

* Line 268: A reference is needed after “Fuller test”.

* Line 275: Do not put a space between “95” and “%”. Simply have “95%”. This occurs frequently throughout the manuscript and needs to be changed in all places.

* Line 290: Add a reference for the “Keras framework”.

* Table 2 is presently unnecessary as you do not talk about runtime as part of your results. This will also be discussed in the “Major” section below.

* Line 299: What do you mean by “preliminary experience”? Describe exactly what this preliminary experience entailed. I.e., what steps did you take to derive the bounds of 10 and 60?

* Line 439: “range” not “rang”.

* Line 462: “Table 6” not “table 6”.

* Line 495: The first five words in this line have no spaces between them. Add the spaces.

* Line 516: “A PSO algorithm” not “An PSO algorithm”.

Major

* Lines 113-121: It would be preferable to study the data from more than one site so that you can compare results and produce more robust results at the end. The results from one site only are not generalisable.

* There is no rationale why the PSO algorithm is chosen to hybridise with LSTM. Perhaps there are better algorithms? Some discussion of this point is necessary.

* Lines 280-281: Why were the data split this way, i.e, different time periods? There may have been different conditions at each of those time periods, making training and validation data sets somewhat incompatible. Why not split it where both sets have the same time period (1980-2018) and use half the data each?

* As you have a hybrid algorithm, i.e., two algorithms are run and co-operate with one another, however, this always incurs a large computational (runtime) cost. Computational times are not reported in this paper (see comment in “Minor” section). How does your hybrid algorithm compare in terms of computational time to the algorithms you report in Table 6? Also, what is the comparison between the two algorithms in your hybrid? What resourcing do each of these required compared to the other? How many iterations are each run for? I see you have 20 for PSO in Line 303, but what are all of the other parameter values and why did you choose the values you used?

Reviewer #2: The paper presents a combination of PSO for selecting values for two hyperparameters for an LSTM ANN for predicting reference evapotranspiration based on meteorological data based on a long time series for a single site. The authors have clearly devoted considerable effort to this, and have in parts conducted rigorous evaluation of the predictor's capabilities. However, there are two major methodological flaws that require the work be reconsidered outside of the review process:

1. The work's stated contribution is in the use of PSO to tune a subset of hyperparameters for LSTM, yet it is not demonstrated why PSO is suitable for this task, when the defined search space is so small (2,550 alternatives). A grid search at different levels of granularity would likely be just as effective at finding good hyperparameter values for equivalent computational effort. (Indeed, given the work only involved three [if I understand correctly] randomised trials the same effort could be used to explore over 45% of the entire search space).

2. The comparison against other techniques for this prediction problem is fatally flawed by comparing reported performance on _different_ datasets. Either rerun your technique on those other datasets or rerun those other approaches on yours (or even do both), but no inferences can be drawn from the current analysis.

Additional comments, mostly in paper order, follow:

PLOS ONE has a very broad audience, so explaining transpiration and evapotranspiration very early in the paper is required. Not everyone will remember their high school biology.

The opening literature review covers a lot but doesn't say much other than that other approaches focus on daily prediction instead of forecasting. I suggest expanding the analysis to better highlight what is novel in your approach.

Description of the approach:

- The components of Figure 3 should be explained in the text. Does it represent a single neuron, or the smallest collection neurons, etc.? Currently a lot of the explanation in the section "Long short-term memory network model" has insufficient detail except for those already familiar with it

- In addition to the detailed explanation given for PSO it would be helpful to describe the communication topology used: this appears to be a gbest PSO (or PSO with a fully-connected topology)

- The gbest PSO is very fast to converge (to local optima). Was this a deliberate design choice? Explain why this version of PSO was chosen

- Another figure illustrating the effect of the different hyperparameters that are controlled by the PSO would greatly help in the explanation of the hybrid system

Additional detail of the hybrid optimisation method is required when it is first described (near the 8 steps):

- Was each PSO particle really only optimising a 2D vector? (See my summary comment (1) at the top)

- The description of the hybrid model implies that the entire population of PSO solutions is used to define just one LSTM network. Presumably that is _not_ the case, but the description currently doesn't make that clear. If it _is_ the case that the population of solutions is used to train a single LSTM network then clearly explain how that is done

- Overall, the 8 steps shown before Figure 4 would be better formatted as an algorithm so that the parts that repeated are clearly distinct from those steps that occur only once at the start or end of the overall process

Regarding the data splits: although contiguous time periods are required for time series regression, is it valid to split the data into time periods that occurred under different climatic conditions? There are clear changes in global ambient temperatures and other meteorological data over the 40-year period in question. While predicting ETo from meteorological data should work for any inputs, splitting the data this way will likely mean training under one set of conditions and then evaluating the model's efficacy under different conditions.

PLOS ONE is a generalist scientific journal, so please include footnotes to explain/link to pandas and keras. (The majority of readers will not know these are Python libraries or what they are for.)

How was each PSO solution's 'fitness value' determined given the four evaluation measures? Was just one used or were they combined in some way?

Issues with the evaluation of the hybrid system:

- Was the hybrid system only run for just three randomised trials? Given the likely variability in such a complex system's behaviour this should be increased to at least 10 to get some representative indication of performance.

- However, given the solution space is 51 (node count range) x 50 (look back range) = 2,550 and each run explores up to 400 (20 solutions x 20 iterations) hyperparameter combinations, which is ~15.5% of the search space, is the PSO really worthwhile? Merely evaluating the system's performance properly with 10 randomised trials would involve the same effort as evaluating 150% of the entire search space. So: is the hybrid system bring presented as a viable alternative for larger problems?

- And: why was a random search of that same search space (also for 400 solutions) not examined to judge the contribution of the PSO search?

It is not clear how the contents of Table 4 were derived. Are these a collection of the best solutions discovered by PSO?

Figure 6 (ETo predictions and true values across 11 years) is of limited utility since it shows not much more than what the numerical accuracy measures did, that the LSTM model does quite a good job and gets it wrong sometimes. Even if the true value (plotted first) was entirely obscured by the predicted value (plotted over the top) then this wouldn't necessarily confirm that the learned model was good, only that its predictions lie somewhere behind the many points plotted over the top.

This text following Figure 6 makes no sense as it describes the 'value range of the prediction residual' twice: The prediction residual of optimized best model for validation set was

376 shown in (Fig 7), it can be seen from the figure that the value range of the 377 prediction residual was from -2.0 to 2.0 mm, and the value range of the 378 prediction residual was from -0.5 to 0.5 mm."

Figure 8: While showing a line of best fit reinforces the linear relationship between predicted and actual values, showing the line representing a perfect relationship would be more meaningful as it reveals that the predictions tend to underestimate higher actual values of ETo, which does not appear to be discussed in the paper.

It is good that the results in Figure 10 are explained in the text, because it is actually not possible to clearly see when the model is predicting higher than ground truth or lower than ground truth. This is because it is difficult to match each predicted point to its associated ground truth point: only some vague trends can be seen. I don't have a good suggestion for this other than the scatter plots you already include of predicted versus actual.

As with Figure 8, Figure 12 would be more informative is the perfect fit line were shown instead, revealing that the model predicts lower for larger ground truth values.

Table 6 should use the same number of significant digits for all entries. And why are the MAE and MSE entries for SVR-RBF not bolded, since they are the same as PSO-LSTM?

Most problematic, why are performance of other models reported for different data within Table 6 (see, e.g., SVR-RBF, RF, CNN and LSTM, which were evaluated for a different location, in Portugal, similarly for the work of Ferreira, which reports performance on data from Brazil, and Farooque, who studied data from PEI [which definitely does _not_ have a monsoonal pattern like the Wugong site you have based your work on]). Such a comparison is completely inappropriate. Those alternative models should be applied to the same training and testing data as used in your work in order to create a valid comparison, or you should apply your approach to those other datasets, or both.

Subset of minor grammatical issues:

- Abstract: opening sentence should be split after irrigation,

- Split the later sentence between "decision, many"

- Inconsistent ET_o formatting (should be fixed during final typesetting)

- Intro, paragraph 1: The final sentence should be split between ET_o and this method

- There are other similar grammatical errors but I will not list any more

- "numbers of" is used where "number of" should be in relation to the number of hidden layers

- Use capital S Section when referring to Section 1, etc.

- In the explanations of the variables in (1) "represent" should be "represents" (third person singular)

- typo line 431 reads 2018 instead of 2019

- typo line 439: rang  range

6. PLOS authors have the option to publish the peer review history of their article (what does this mean?). If published, this will include your full peer review and any attached files.

Reviewer #1: **Yes: **Marcus Randall

Reviewer #2: **Yes: **James Montgomery

---

## [Author Response · Author response to Decision Letter 0]

20 Sep 2022

Dear Editors and reviewer,

 Thank you for your giving us an opportunity to revise our manuscript! Now we are submitting the revised manuscript entitled “Daily reference evapotranspiration prediction for irrigation scheduling decisions based on the hybrid PSO-LSTM model” for consideration for publication in PLOS ONE.

 It can be found that the prediction accuracy of the optimized model in the revised manuscript is lower than the prediction accuracy of the model in the original manuscript. The main reason is that the model is overtrained in the original manuscript. Conversely, the optimized model in the revised manuscript is not overfitted.

In the revision, we have performed nearly all the suggested experiments, data analyses, and fully addressed the comments made by the reviewers and editor.

---

## [Decision Letter · Decision Letter 1]

18 Nov 2022

PONE-D-22-09180R1Daily reference evapotranspiration prediction for irrigation scheduling decisions based on the hybrid PSO-LSTM modelPLOS ONE

Dear Dr. Jia,

Thank you for submitting your manuscript to PLOS ONE. After careful consideration, we feel that it has merit but does not fully meet PLOS ONE’s publication criteria as it currently stands. Therefore, we invite you to submit a revised version of the manuscript that addresses the points raised during the review process. As both reviewers have noted, the significant efforts made to respond to reviewers' comments have been recognised. One reviewer has noted that a case for using PSO has still not been made. Stating the mechanism of the algorithm's operation does NOT justify its choice - it is important to present at least some reasoning as to WHY this particular algorithm was considered necessary for this problem. I would urge careful consideration of this and related points, as noted in comments 1 and 1.2 in particular. Responding to this issue is critical. Consideration should also be given to the comments regarding the argument presented starting line 364, and whether it is needed. A number of other, minor comments could also be addressed to tidy up the manuscript.

We look forward to receiving your revised manuscript.

Kind regards,

Andrew Lewis

Academic Editor

PLOS ONE

Journal Requirements:

Reviewers' comments:

Reviewer's Responses to Questions

**Comments to the Author**

1. If the authors have adequately addressed your comments raised in a previous round of review and you feel that this manuscript is now acceptable for publication, you may indicate that here to bypass the “Comments to the Author” section, enter your conflict of interest statement in the “Confidential to Editor” section, and submit your "Accept" recommendation.

Reviewer #1: All comments have been addressed

Reviewer #2: (No Response)

2. Is the manuscript technically sound, and do the data support the conclusions?

Reviewer #1: Yes

Reviewer #2: Partly

3. Has the statistical analysis been performed appropriately and rigorously? 

Reviewer #1: Yes

Reviewer #2: Yes

4. Have the authors made all data underlying the findings in their manuscript fully available?

Reviewer #1: No

Reviewer #2: Yes

5. Is the manuscript presented in an intelligible fashion and written in standard English?

Reviewer #1: Yes

Reviewer #2: Yes

6. Review Comments to the Author

Reviewer #1: I appreciate the authors' time and effort in the revision of this manuscript. The authors have done their best to address my comments.

Reviewer #2: I'd like to thank the authors for responding so positively to the critiques of the original submission. As with that original submission it is clear that a significant amount of researcher and computer time has been invested in the work. The expanded number of datasets and alternative ways they have been split to create different training and testing combinations is a really welcome change to the study.

There remain two issues with the work, one of which has not been addressed from the original reviews.

1. While it's good that the runtime is now given (see major point 2 below), and in part this explains the low number of solutions produced by the solver and, the question remains: why PSO? No arguments are put forward to explain why such a metaheuristic will give better search performance than some other (simpler) alternative, and certainly no experimental results are included to demonstrate this either. Some specific issues:

1.1 Figures 7 and 11 show that the minimum MAE of solutions at the end of the run is similar to the minimum at the start. Somewhat worringly, the minimum value is not monotonically decreasing, which should not be possible if particles' pbest solutions are shown. As in my original review: what has PSO added that random search could not achieve given equivalent computational effort?

1.2 The long runtime notwithstanding the argument for using PSO has still not been made. The responses to reviewer comments do not justify its use (most merely re-explain this now very well-known algorithm) and instead repeatedly point out the long time to train the models. What evidence is there that (1) any form of metaheuristic search and (2) PSO in particular is more suitable for this problem than using some other technique (random search, grid search at varying granularities) with the same budget of 400 solution evaluations? Some attempt to address this question needs to be made. (Note that pointing to the improvement in solution quality over time does _not_ justify the use of PSO; because it is elitist the quality of each solution's pbest will be monotonically improving over time.)

1.3 (minor) It would still be worth explaining _why_ a gbest PSO was selected. This only requires a sentence and could just focus on the small number of iterations required by the long time to evaluate each solution.

2. It is good that the training time for the LSTM (hence, a measure of solution evaluation time) is now clearly stated, as this makes the number of iterations used much clearer. However, is this long runtime just an artefact of running it on an underpowered CPU (an i5 is only suitable for running office applications) and not a GPU as keras is intended? Using a more appropriate experimental machine would likely allow for more thorough (and far less time-consuming) experimental work.

Other issues/comments:

- There is a conflict between the argument put forward for why the effect of climate change can be ignored in the data and the fact that alternative splits that use different time periods (past v future, future v past) have been used; the use of additional data splits is otherwise a really good inclusion in the work. The argument that begins line 364 begins by acknowledging the change in meteorological data over the 40 years, but it is nonsensical to compare the magnitude of the standard deviations of measures that use different units to argue that changes to the input features can be ignored. Further, comparing descriptive statistics over the entire time period cannot be used to dismiss observable differences within different contiguous subsets of the data. Is the argument starting at line 364 necessary given many of the experiments are made with different temporal splits? If you believe so then a _much_ stronger argument is required.

- Minor: the new text describing keras as 'an API designed for human beings, not machines' is marketing spin and does not belong in a scientific paper.

- In Section 2.1 Tables 2 and 4 are going to be difficult to format for the journal. However, it's also very difficult for the reader to interpret. Wouldn't these be better presented graphically?

- The discussion of performance based on data split (around lines 548-9) is odd/uninformative. Since the different data splits correspond to training under one climate condition and testing on another, can something more be said about this outcome?

Minor typos noticed while reading:

- General: Please re-check the formatting after applying the changes, as some necessary spaces and text are missing and additional spaces have been inserted where they should not be.

- line 94: the edits have deleted the start of the sentence that used to say PSO optimizes...

- line 124: missing spaces after each of the place names and the opening (. See general comment above as that is not the only problem with spacing in this revised text

7. PLOS authors have the option to publish the peer review history of their article (what does this mean?). If published, this will include your full peer review and any attached files.

Reviewer #1: **Yes: **Marcus Randall

Reviewer #2: No

---

## [Author Response · Author response to Decision Letter 1]

2 Jan 2023

Dear reviewer,

 Thank you for your giving us an opportunity to revise our manuscript again! 

Reviewer #2: I'd like to thank the authors for responding so positively to the critiques of the original submission. As with that original submission it is clear that a significant amount of researcher and computer time has been invested in the work. The expanded number of datasets and alternative ways they have been split to create different training and testing combinations is a really welcome change to the study.

There remain two issues with the work, one of which has not been addressed from the original reviews.

1.While it's good that the runtime is now given (see major point 2 below), and in part this explains the low number of solutions produced by the solver and, the question remains: why PSO? No arguments are put forward to explain why such a metaheuristic will give better search performance than some other (simpler) alternative, and certainly no experimental results are included to demonstrate this either. Some specific issues:

Response: in order to compare the search performance of metaheuristic and simpler alternative, the grid search method were used to optimize the hyper parameters of the two LSTM models, the grid search schemes as show in Table 1. For the first LSTM model, the look back (T1) is increase from 12 to 50, at the same time, the first hidden layer neurons (N1) is increase from 45 to 60, a total of 608 hyperparameters combinations were trained, and the hyperparameter combination corresponding to the minimum fitness is [13, 52], it means that the look back =13, the number of first layer neurons=52 is the optimal combination for the first model. 

For the second LSTM model, the look back (T2) is 5, 50, 100, 150, and 200, respectively. The first and second hidden layer neurons (N2 and N3) are increase from 5 to 200 by 5 each time, a total of 9000 hyperparameters combinations were trained, and the hyperparameter combination corresponding to the minimum fitness is [200, 140, 170, 0.1], it means that the look back are 200, the number of first and second layer neurons are 140 and 170, respectively, and the Dropout are 0.1 are the optimal combination for the second model.

Table 1. The two grid search schemes for two model parameters.

Model Look back First hidden neurons Second hidden neurons Number of trained models

#1 [12,13,14,···,50] [45, 46, 47,···,60] / 608

#2 [5,50,100,150,200] [5, 10, 15,···,200] [5, 10, 15, ···,200] 9000

The table 2 shows the results of comparison of minimum fitness and cost time between grid search and PSO-LSTM. For the first model, the minimum fitness of two methods are 0.595 and 0.597, respectively, the trained time are 58.33h and 221.25h, respectively. It can be seen that the difference between the optimal minimum fitness of the two methods is small, and the cost time of grid search method is far less than that of the PSO method. This means that for the first LSTM model, the grid search method is a better model hyper parameters optimization method.

 For the second model, the minimum fitness of two methods are 0.593 and 0.600, respectively, the trained time are 4826.94h and 806.03h, respectively. It can be seen that the difference between the optimal minimum fitness of the two methods is also small, and the cost time of PSO method is far less than that of the grid search method. This means that for the second LSTM model, the PSO method is a better model hyper parameters optimization method.

Table 2. Comparison of minimum fitness and cost time between Grid search and PSO-LSTM.

Model Minimum fitness (MAE) Cost time (h)

 Grid search PSO Grid search PSO

#1 0.595 0.597 58.33 221.25

#2 0.593 0.600 4826.94 806.03

In general, the grid search method is suitable for the LSTM model with few hyper parameters, for the LSTM models with multiple hyper parameters and a large range of hyper parameters, the training time cost of grid search method is very high. It can be seen that the PSO method can not only optimize multiple parameters in a relatively small amount of time, but also has a relatively high optimization accuracy, therefore, the PSO algorithm is more suitable for optimizing the LSTM model with multiple hyper parameters. 

　　　　　　　　　

1.1Figures 7 and 11 show that the minimum MAE of solutions at the end of the run is similar to the minimum at the start. Somewhat worringly, the minimum value is not monotonically decreasing, which should not be possible if particles' pbest solutions are shown. As in my original review: what has PSO added that random search could not achieve given equivalent computational effort?

Response: in this paper, the grid search method and PSO algorithm were used to optimized two type different LSTM models, and more than 10 000 models were trained and optimized. It is undeniable that the MAEs of those LSTM models have little difference. So the Figure 7 and 11 also show that the minimum MAE of solutions at the end of the run is similar to the minimum at the start. Considering the unpredictability of hyper parameters of the model, the minimum is not absolutely monotonically decreasing. 

1.2The long runtime notwithstanding the argument for using PSO has still not been made. The responses to reviewer comments do not justify its use (most merely re-explain this now very well-known algorithm) and instead repeatedly point out the long time to train the models. What evidence is there that (1) any form of metaheuristic search and (2) PSO in particular is more suitable for this problem than using some other technique (random search, grid search at varying granularities) with the same budget of 400 solution evaluations? Some attempt to address this question needs to be made. (Note that pointing to the improvement in solution quality over time does _not_ justify the use of PSO; because it is elitist the quality of each solution's pbest will be monotonically improving over time.)

1.3 (minor) It would still be worth explaining _why_ a gbest PSO was selected. This only requires a sentence and could just focus on the small number of iterations required by the long time to evaluate each solution.

Response: The PSO is the only form of metaheuristic search for the LSTM model in this paper, In fact, many swarm intelligence algorithms were used to optimize the hyperparameters of deep learning network, more and more relevant research literature are published. The Whale optimization Algorithm (WOA) and Grey Wolf Optimizer (GWO) were also used to optimized the LSTM model, and these two optimization algorithms also take a long time, and the optimization models are easy to fall into local optimization, so this paper dose not show the optimized result of these two models. If necessary, the optimization results of the these two models can also be provided.

One of the most common approach for optimizing the hyperparameters is through grid search. However, grid search suffers the curse of dimensionality when the number of hyperparameters are very large. Based on the reviewers’ suggestions, the grid search were also used to optimized the LSTM models, and 9000 LSTM models were trained, the best LSTM network structure has been selected, the results show that that the training time of the grid search is more longer, but the optimization results of the grid search algorithm are not significant. The random search algorithm has not been trained in the paper. The PSO algorithm has the advantages of a simple structure, easy implementation, and a strong global capability, which seeks the optimal value of the complex space through information transmission and competition between individuals. 

In general, compared with the GWO, WOA and the grid search algorithms, the PSO algorithms is a relatively good algorithm.

2. It is good that the training time for the LSTM (hence, a measure of solution evaluation time) is now clearly stated, as this makes the number of iterations used much clearer. However, is this long runtime just an artefact of running it on an underpowered CPU (an i5 is only suitable for running office applications) and not a GPU as keras is intended? Using a more appropriate experimental machine would likely allow for more thorough (and far less time-consuming) experimental work.

Response: Thanks a lot for the reviewer’s suggestions. Using a more appropriate experimental machine would likely allow for more thorough experimental work, in order to achieve the comparability of different optimization methods, a more appropriate experimental machine was not used. To improve the training efficiency of the LSTM model, the grid search optimization LSTM were trained on four computers with the same configuration, and the grid search algorithm has been running on four computers for 4826.94h.

Other issues/comments:

- There is a conflict between the argument put forward for why the effect of climate change can be ignored in the data and the fact that alternative splits that use different time periods (past v future, future v past) have been used; the use of additional data splits is otherwise a really good inclusion in the work. The argument that begins line 364 begins by acknowledging the change in meteorological data over the 40 years, but it is nonsensical to compare the magnitude of the standard deviations of measures that use different units to argue that changes to the input features can be ignored. Further, comparing descriptive statistics over the entire time period cannot be used to dismiss observable differences within different contiguous subsets of the data. Is the argument starting at line 364 necessary given many of the experiments are made with different temporal splits? If you believe so then a _much_ stronger argument is required.

Response: the ETo was calculated by mixing six weather parameters, and the change in meteorological data over the 40 years are obvious, therefore, the ETo in the past 40 years has also changed. The research interval in this paper is 1 day, and the change of meteorological parameters in 1 day is random. This paper cannot accurately describe the change of meteorological parameters, so this paper ignores the influence of meteorological parameter changes on the prediction model. The data were split into training set (1980-2007), test set (2008-2018) and prediction set (2019), the data number of training set, test set and prediction set were 10227, 4021, and 365, respectively. 

- Minor: the new text describing keras as 'an API designed for human beings, not machines' is marketing spin and does not belong in a scientific paper.

Response: the reference 35 were deleted, and revised to [35] Yulius Harjoseputro. A Classification Javanese Letters Model using a Convolutional Neural Network with KERAS Framework[J]. International Journal of Advanced Computer Science and Applications (IJACSA),2020,11(10)..10.14569/IJACSA.2020.0111014.

- In Section 2.1 Tables 2 and 4 are going to be difficult to format for the journal. However, it's also very difficult for the reader to interpret. Wouldn't these be better presented graphically?

Response: The format of Table 2, 3, 4 and 5 have been modified in the revised manuscript.

-The discussion of performance based on data split (around lines 548-9) is odd/uninformative. Since the different data splits correspond to training under one climate condition and testing on another, can something more be said about this outcome?

Response: the different data splits correspond to training under one climate condition, and in order to verify the accuracy of the optimized model under different period, four types of periods are divided, including type A (training set 1980-2007,10220, test set 2008-2018, 4015), type B (training set 1980-2017,13869, test set, 1981-2018,13869), type C (training set 2003-2014, 4380, test set, 2015-2018,1460) and type D (training set 2011-2017, 2190, test set, 2017-2018, 730). The optimized model were also trained and tested under four different datasets, it must be pointed out that the optimized model is retrained and tested with four different data sets again, instead of just using the optimization model to predict different test sets.

Minor typos noticed while reading:

- General: Please re-check the formatting after applying the changes, as some necessary spaces and text are missing and additional spaces have been inserted where they should not be.

- line 94: the edits have deleted the start of the sentence that used to say PSO optimizes...

- line 124: missing spaces after each of the place names and the opening (See general comment above as that is not the only problem with spacing in this revised text.

Response: The above minor revisions have been completed in the revised manuscript.

At the same time, the results of grid search method for two models were added in the supporting information.

All authors have read and approved the re-submission of the manuscript!

If you have any question, please let me know!

Thank you for your consideration of our paper and we are looking forward to hearing from you!

Sincerely, Weibing Jia，jiawb02@stu.xjtu.edu.cn

PhD student，The State Key Laboratory for Manufacturing Systems Engineering, Xi’an Jiaotong University，Xi’an 710049, China

---

## [Editor Report · Decision Letter 2]

25 Jan 2023

Daily reference evapotranspiration prediction for irrigation scheduling decisions based on the hybrid PSO-LSTM model

PONE-D-22-09180R2

Dear Dr. Jia,

We’re pleased to inform you that your manuscript has been judged scientifically suitable for publication and will be formally accepted for publication once it meets all outstanding technical requirements.

Kind regards,

Andrew Lewis

Academic Editor

PLOS ONE

Additional Editor Comments (optional):

The point raised by the reviewers' regarding the justification of the choice of PSO as the optimising algorithm seems mostly to have been addressed by post hoc arguments. It is not a matter of whether PSO is not an appropriate or well-suited algorithm for the purposes of this problem. As a matter for future consideration, when choosing an algorithm for a particular problem, we should think beforehand about what the problem needs, which of the many different algorithms and approaches possible will be most effective and efficient in meeting those needs, and explain why.
---

## [Editor Report · Acceptance letter]

10 Apr 2023

PONE-D-22-09180R2 

Daily reference evapotranspiration prediction for irrigation scheduling decisions based on the hybrid PSO-LSTM model 

Dear Dr. Jia:

I'm pleased to inform you that your manuscript has been deemed suitable for publication in PLOS ONE. Congratulations! Your manuscript is now with our production department. 

Kind regards, 

on behalf of

Dr. Andrew Lewis 

Academic Editor

PLOS ONE